# Estimating HIV incidence and assessing associated risk factors among adults: Evidence from the 2018–2022 HIV vaccine preparedness cohort in Masaka, Uganda

Sylvia Kusemererwa[1]*, Sheila Kansiime[1,2], Gertrude Mutonyi[1], Ayoub Kakande[1], Shamim Nabukenya[1], Aeron Namirembe[1], Naphtal Erima[1], Doreen Joan Asio[1], Elizabeth Mbabazi Atuhura[1], Shamim Ssendagire[1], Martin Onyango[1], Julie Fox[3], Sheena McCormack[4], Pontiano Kaleebu[1,2], Eugene Ruzagira[1,2], on behalf of the PrEPVacc Study Team¶

1 Medical Research Council/Uganda Virus Research Institute and London School of Hygiene and Tropical Medicine, Uganda Research Unit, Entebbe, Uganda, 2 London School of Hygiene and Tropical Medicine, London, United Kingdom, 3 King's College London, London, United Kingdom, 4 Medical Research Council Clinical Trials Unit, London, United Kingdom

¶ PrEPVacc Study Team is provided in the Acknowledgments.
* sylvia.kusemererwa@mrcuganda.org, kansykarus@gmail.com

## Abstract

### Introduction

Despite global reports of declining HIV incidence, current data are limited. This study presents findings on HIV incidence and risk factors in a HIV trial preparedness cohort.

### Methods

Between 18 July 2018 and 13 October 2022, individuals 18–45-year-old were recruited from communities along the trans-African highway and the shores of Lake Victoria in Masaka district, Uganda. Eligible individuals were HIV-negative and met at least one of the following criteria: suspected or confirmed sexually transmitted infection (STI), unprotected sex with ≥2 partners, unprotected sex with a new partner in the past 3 months, or unprotected sex in exchange for money/goods in the past month. Baseline data included demographics, sexual behaviour, and HIV risk factors. Follow-up assessments of HIV risk and sexual behaviour were conducted every six months, while HIV counselling and testing (HCT) were provided every 3 months, with linkage to care for those testing positive. Data were summarised descriptively, and associations with HIV incidence were analysed using uni-variable and multi-variable Poisson regression models.

**Data availability statement:** The datasets generated and analysed during this study are available through the London School of Hygiene & Tropical Medicine (LSHTM) Data Compass at the following DOI: https://doi.org/10.17037/DATA.00004608 [50]. These data are managed by the MRC/UVRI and LSHTM Uganda Research Unit and are available in accordance with their data sharing policy and procedures, outlined at https://apps.mrcuganda.org/mrcdatavisibility/Home/DataShare. Due to ethical considerations and the commitment to protect participant confidentiality, access to the fully anonymised dataset is controlled. Data are available for research verification purposes upon reasonable request. Interested researchers must submit an application including a description of the intended research, requested data variables, and any other relevant details. Applications are reviewed by the study team at MRC/UVRI and LSHTM Uganda Research Unit, in consultation with external experts such as ethics representatives. If the proposed use is approved under the study's ethical conditions, access will be granted under a Data Transfer Agreement, which requires that data be stored securely and used solely for the approved purpose.

**Funding:** The PrEPVacc registration cohort study was funded by The second European & Developing Countries Clinical Trials Partnership (EDCTP2) (Grant number: RIA-2016V-1644).

**Competing interests:** The authors have declared that no competing interests exist.

## Results

Of the 1422 individuals enrolled, 1115 (78.4%) attended ≥1 follow-up visit; 69% were female and 55% were aged ≤24 years. Over 900.3 person-years of observation (PYO), 24 individuals acquired HIV, yielding an incidence rate (IR) of 2.7 PYO [95% Confidence Interval (CI): 1.8–4.0]. Baseline factors independently associated with incident HIV included female sex [adjusted incidence rate ratio (aIRR) = 6.84 PYO, 95% CI 1.60–29.30], residence in a fishing community [aIRR = 3.04 PYO, 95% CI 1.05–8.78] and recreational drug use in the past 3 months [aIRR = 3.08 PYO, 95% CI 1.19–7.99]. In a separate analysis, using time-updated variables assessed during follow-up, HIV acquisition was associated with sex after alcohol consumption [aIRR = 2.65 PYO, 95% CI 1.11–6.31], and STI diagnosis/treatment within the past 3 months [aIRR = 2.52 PYO, 95% CI 1.09–5.80].

## Conclusion

HIV incidence remains high among women, fishing community residents, and individuals with high risk behaviours, underscoring the need for targeted interventions to reduce the burden in these populations.

## Introduction

Human immunodeficiency virus (HIV) continues to be a significant global health challenge despite decades of research and intervention efforts. In 2023, there were 1.3 million new infections, with about half occurring in sub-Saharan Africa [1]. A substantial proportion (55%) of these infections were among key populations and their partners [2], including men who have sex with men, female sex workers, injecting drug users, and others. HIV prevention research, particularly on interventions like vaccines, focus on enrolling key populations at high risk of HIV acquisition to assess efficacy [3,4]. The success of these trials depends on factors such as HIV incidence, participant accrual, duration of participant follow-up, and maximising retention [5]. However, in the past, studies were prematurely terminated due to unexpectedly low HIV incidence leading to insufficient statistical power [6].

Global HIV incidence is declining, with fewer new infections reported in 2023 than at any point since the mid-1990s [1]. The most significant reductions have occurred in sub-Saharan Africa, where new infections among adults aged 15–49 dropped by 54%, from 1.1 million in 2010–510,000 in 2022 [2]. This decline is largely attributed to the expansion of prevention strategies such as oral pre-exposure prophylaxis (PrEP) and universal test-and-treat programs [7,8]. However, current incidence data remain limited. A 2021 systematic review of data from the reported HIV incidence rates ranging from 0.6 to 3.4 per 100 person-years (PYO) among sex workers, 1.03 to 15.4 per 100 PYO among men who have sex with men, and 1.59 to 6.04 per 100 PYO among fisher folk [9]. While declining incidence reflects progress in HIV prevention, it also poses challenges for HIV prevention trials by reducing observable new infections,

lowering statistical power, and increasing trial complexity and costs [4,10–13]. Addressing these challenges requires larger sample sizes, longer follow-up periods, and the integration of proven prevention methods with new interventions [14]. To improve trial efficiency, researchers incorporate consideration of pre-existing infections, product adherence, and discontinuation in sample size calculations, while also developing biomarkers, and alternative trial designs [15]. Consequently, relative efficacy, rather than absolute efficacy, has becomes a key measure of intervention success [16]. Given the evolving HIV prevention landscape, updated incidence data are essential—particularly in the era of PrEP—to ensure effective trial planning.

To support the PrEPVacc HIV vaccine efficacy trial [17], we conducted the PrEPVacc registration cohort study across five research sites in Mozambique, South Africa, Tanzania, and Uganda. This study aimed to recruit high-risk, HIV-negative individuals, estimate HIV incidence, and inform sample size calculations. In Uganda, the study took place in Masaka, a hub for HIV research. Previous vaccine preparedness studies in Masaka reported HIV incidence rates ranging from 1% in rural communities to 4% among serodiscordant couples [18]. Additionally, earlier research estimated an incidence of 4.9 cases per 100 PYO among high-risk fishing communities [19]. Key risk factors for HIV acquisition included female sex, marital status, multiple sexual partners, a history of sexually transmitted infections (STIs), and lack of circumcision [20,21]. Further studies linked younger age, alcohol use, and genital discharge to increased risk among serodiscordant couples [22]. Given that these data are outdated, updating HIV incidence estimates and risk factors was essential for a more accurate and current assessment. This paper presents findings on HIV incidence and associated risk factors among participants in Masaka, offering critical insights to guide future HIV prevention research.

## Materials and methods

### Study design

Longitudinal prospective cohort.

### Study setting

The PrEPVacc registration cohort study was conducted between 18 July 2018 and 13 October 2022. The main aim of the cohort study was to identify HIV-negative adults at high risk of HIV infection and prepare them for participation in a phase IIb HIV prophylactic vaccine trial (NCT04066881) at sites in Mozambique, South Africa, Tanzania, and Uganda. The trial aimed to determine the efficacy of two HIV prophylactic vaccine regimens and to compare the effectiveness of Truvada and Descovy as PrEP [17].

In Uganda, the cohort study was conducted at the Medical Research Council/Uganda Virus Research Institute and London School of Hygiene and Tropical Medicine (MRC/UVRI and LSHTM) Uganda Research Unit's clinical research site in Masaka city, Masaka district, located 120 kilometres southwest of the capital, Kampala.

### Study population and recruitment

Potential study participants were recruited from 10 communities along the trans-African highway and 13 fishing villages along the shores of Lake Victoria, within a radius of approximately 80 km from Masaka city. Eligible individuals were required to be 18–45 years old, HIV-negative, willing to provide locator information and available for follow-up and be at risk of HIV infection. Risk criteria included suspected/confirmed sexually transmitted infection (STI), unprotected sex with ≥2 partners, unprotected sex with a new partner in the past 3 months, or unprotected sex in exchange for money/goods in the past month.

Prospective volunteers were identified through HIV counselling and testing (HCT) outreaches in the study communities and through a door-to-door approach. Study HIV counselors, supported by community mobilizers, provided HCT, and collected information on age and HIV risk behaviour. Individuals who tested HIV-negative were provided brief information about the study and those who expressed interest invited to the study clinic for screening and possible enrollment.

## Procedures

Screening and enrolment were conducted at the same visit. Screening procedures included providing detailed study information, obtaining written informed consent, repeat HCT, urine pregnancy testing (women), eligibility assessment, and enrolment for those that were found to be eligible. Procedures at enrolment included collection of locator information, socio-demographic data, HIV risk and risk reduction behaviours (including awareness and use of oral PrEP) data, provision of information and counselling on PrEP and for those willing to initiate PrEP, referral to a PrEP provider.

All volunteers were followed quarterly. At these visits, locator information was reviewed, updated and HCT performed. Urine pregnancy testing (women) and collection of data on HIV risk and risk reduction behaviour were performed every 6 months. Participants who tested HIV positive at any visit were immediately referred for HIV care and treatment at HIV care centres of their choice. Women who tested positive for pregnancy were referred for antenatal care services at a health facility of their choice. Additionally, pregnant women who tested HIV positive were referred for prevention-of-mother-to-child services.

Participants were required to attend at least one follow-up visit before being screened for the PrEPVacc HIV vaccine trial and could be followed for up to three years. Those found ineligible for the vaccine trial but eligible to continue in the registration cohort were expected to remain in the study for a minimum of 12 months from their enrolment date. A participant completed study follow-up when: i) they enrolled in the PrEPVacc trial, ii) withdrew consent, iii) tested HIV-positive, iv) were determined to be ineligible for continued follow-up based on a re-assessment of his/her risk and study eligibility or, v) were non-compliant to the study procedures or when the PrEPVacc trial was fully enrolled. All study procedures were conducted at the research site.

To ensure data quality, study staff were trained in good clinical practice, data protection regulations, study protocol, the operations manual, standard operating procedures, and the data collection tools. Data were routinely cleaned and managed using automated checks at entry to identify missing or inconsistent values. In addition, periodic data quality checks were conducted using Stata, and any queries generated were verified and resolved by research staff through review of source documents. Where necessary, participants were re-contacted to clarify or complete missing information.

## Laboratory procedures

HIV testing was conducted according to the national HCT guidelines [23,24]. Blood obtained by venipuncture was tested using HIV rapid test kits: Alere Determine HIV1/HIV-2 (Alere Medical Co Ltd, 357 Matsuhidai Matsudo-shi, Chiba-ken 270-2214, Japan) for screening, Stat-Pak HIV 1/2 (Chembio Diagnostic systems, New York, NY11763, USA) for confirmation of positive results, and SD Bio line (Standard Diagnostics, Kyonggi-do, South Korea) as a tie-breaker. The screening algorithm has a sensitivity of 99.2%, specificity of 99.1%, and positive predictive value of 99.0% and negative predictive value of 99.2% [25].

A volunteer was considered HIV-positive if two antibody rapid tests were positive (S1 Fig). Urine was tested for pregnancy using ßhCG reagent strips (QuickVue hCG Combo, Quidel Cooporation, San Diego, CA92121, USA).

To ensure the quality HIV testing procedures, certified HIV testing service counsellors performed HCT. Internal quality control (IQC) procedures were performed weekly using known positive and negative samples, while external quality assurance (EQA) assessments were conducted quarterly under the Uganda Virus Research Institute's national reference laboratory EQA proficiency testing scheme. For urine pregnancy testing, IQC procedures were performed daily before testing participant samples using Bio-Rad urinalysis controls (Level 1 and Level 2), and EQA was conducted through the Royal College of Pathologists of Australia (RCPA) program.

## Statistical analysis

Data were managed using Open Clinica (community edition) and analysed in Stata version 18.0 (College Station, TX, US). HIV incidence was calculated as the number of seroconversions divided by the total follow-up time, measured in

person-years. Follow-up time was defined as the period from enrolment to the last HIV test for participants who remained HIV-negative or to the first positive test for those who seroconverted. Associations with HIV incidence were assessed using univariable and multivariable Poisson regression, with the natural log of person-time (years) included as an offset.

Two primary multivariable analyses were conducted: (i) using demographic and behavioural risk data collected at baseline, and (ii) using behavioural risk data from the last available follow-up visit, adjusted for potential confounders identified in the first analysis.

In the univariable analysis, factors were selected based on existing literature demonstrating associations with HIV incidence and their availability in the study dataset. These included sociodemographic characteristics (age, gender, occupation, education, marital status, and residence in a fishing village versus non-fishing village) and behavioural HIV risk indicators such as condom use, transactional sex in the last 3 months, age difference with sexual partners, recreational drug use, sex after alcohol consumption, number of partners, and STI-related symptoms (e.g., abnormal genital discharge, ulceration) or self-reported STI diagnoses in the last 3 months. The follow-up period (first, second, or third year) and the calendar period during which PYO were accrued, were also assessed.

In the two multivariable analyses, all factors of interest were initially included based on the univariable analysis results. Factors with a P-value <0.2 were retained using backward elimination based on the Likelihood Ratio test. Age and sex were included *a priori* in all models based on evidence from previous research [26–28].

Furthermore, a supplementary analysis using mixed-effects Poisson regression models with individual-specific random effects was conducted to account for repeated assessments over time and investigate their associations with HIV acquisition.

### Ethical considerations

The study was approved by the Uganda Virus Research Institute Research Ethics Committee (GC/127/18/03/637), the Uganda National Council for Science and Technology (HS2392), and London School of Hygiene and Tropical Medicine Ethics Committee (26494). All participants provided written informed consent before undergoing study procedures. An impartial witness was present to oversee the consent process for illiterate participants. Those willing to initiate oral PrEP were referred to a provider of their choice. Individuals who tested HIV-positive at any of the study visits were provided post-test counselling and linked to HIV care.

## Results

### Baseline characteristics

A total of 1,736 adults were screened, of whom 1,422 (82%) were enrolled. The most common reason for screen failure (n = 314) was low risk of HIV acquisition, accounting for 94% (n = 295) of the failures. Among those enrolled, 1,115 (78%) attended at least one follow-up visit. of these, 691 (62%) exited the study early, mainly due to declining further follow-up (30%, n = 206) or being re-classified as low risk for HIV acquisition (21%, n = 142). Overall, 424 (38%) participants completed the study (Fig 1). Participants who did not attend any follow-up visits were more likely than those who did to be female (93% vs 69%, p-value<0.001), sex workers (69% vs 46%, p-value<0.001), and residents of non-fishing villages (94% vs 83%, p-value<0.001), (S1 Table).

Among participants who were enrolled and followed up, 69% (n = 769) were female and 55% (n = 612) were aged 24 years or younger. Most participants (61%) had primary school education or lower, 53% were single, and 46% were sex workers. The majority (83%) resided in non-fishing villages (Table 1).

### HIV incidence and baseline risk factors

Overall, 24 individuals acquired HIV during 900.3 PYO, resulting in an incidence rate (IR) of 2.7/100 PYO [95% Confidence Interval (CI): 1.8–4.0]. HIV incidence was highest among female participants [IR = 4.2/100 PYO, 95% CI 2.7–6.7],

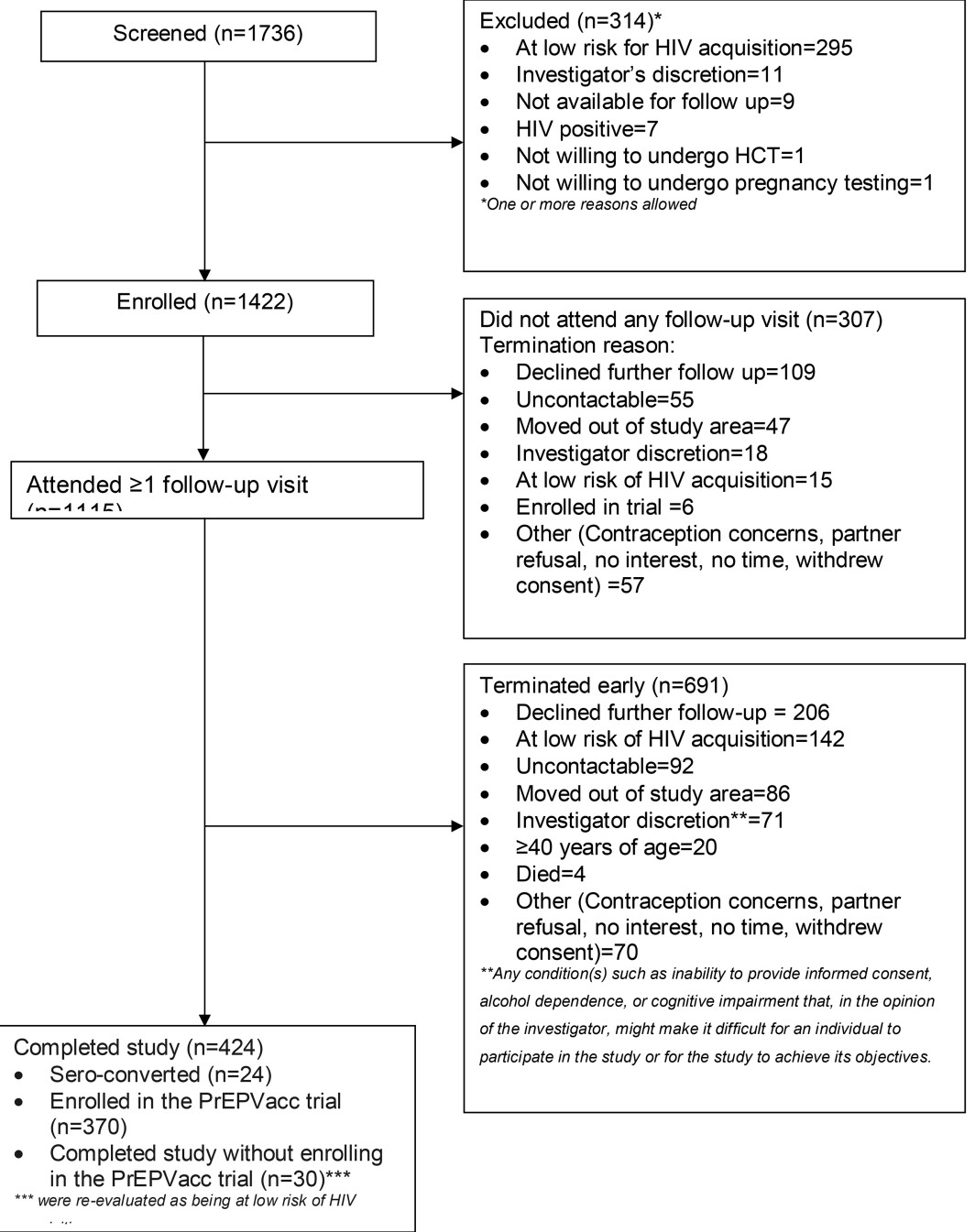

**Fig 1.  Study profile of adults screened and enrolled into the vaccine preparedness cohort in Masaka, Uganda (July 2018-October 2022).**

those aged 24 years or younger [IR = 3.2/100 PYO, 95% CI 1.9–5.3], hair salon/bar/lodge workers [IR = 5.8/100 PYO, 95% CI 3.1–10.8], participants with primary school education or lower [IR = 3.0/100 PYO, 95% CI 1.8–4.8], divorced/separated/widowed individuals [IR = 3.4/100 PYO, 95% CI 1.3–9.1], and those who reported use of recreational drugs in the past 3 months [IR = 4.8/100 PYO, 95% CI 2.3–10.2] (Table 1).

**Table 1. Baseline factors associated with HIV acquisition among adults enrolled in an HIV vaccine preparedness study in Masaka, Uganda.**

| Baseline characteristics | Number of participants (%) | HIV incidence | | | Univariable analysis | | Multi-variable analysis | |
|---|---|---|---|---|---|---|---|---|
| | | N | PYO | IR/100PYO | IRR (95% CI) | P-value | aIRR (95% CI) | P-value |
| Overall | 1115 (100) | 24 | 900.3 | 2.7 (1.8–4.0) | | | | |
| **Gender** | | | | | | | | |
| Male | 346 (31) | 5 | 452.6 | 1.1 (0.5–2.7) | **Ref** | | **Ref** | |
| Female | 769 (69) | 19 | 447.7 | 4.2 (2.7–6.7) | 3.84 (1.43–10.29) | 0.007 | 6.84 (1.60–29.30) | 0.010 |
| **Age** | | | | | | | | |
| ≤24 | 612 (55) | 15 | 467.9 | 3.2 (1.9–5.3) | 1.54 (0.67–3.52) | 0.306 | 1.33 (0.55–3.20) | 0.522 |
| >24 | 503 (45) | 9 | 432.4 | 2.1 (1.1–4.0) | **Ref** | | **Ref** | |
| **Occupation** | | | | | | | | |
| Other§ | 337 (30) | 3 | 366.8 | 0.8 (0.3–2.5) | **Ref** | | **Ref** | |
| Sex worker | 509 (46) | 7 | 209.0 | 3.3 (1.6–7.0) | 4.10 (1.06–15.84) | | 1.25 (0.24–6.43) | |
| Salon/Lodge/Bar worker | 181 (16) | 10 | 171.4 | 5.8 (3.1–10.8) | 7.13 (1.96–25.92) | | 3.51 (0.83–14.84) | |
| Subsistence fisheries worker | 88 (8) | 4 | 153.1 | 2.6 (1.0–7.0) | 3.19 (0.71–14.26) | 0.026 | 2.75 (0.54–14.09) | 0.139 |
| **Education level** | | | | | | | | |
| ≤Primary | 677 (61) | 17 | 572.8 | 3.0 (1.8–4.8) | **Ref** | | | |
| ≥Secondary | 438 (39) | 7 | 327.5 | 2.1 (1.0–4.5) | 0.72 (0.30–1.74) | 0.465 | | |
| **Marital status** | | | | | | | | |
| Single | 595 (53) | 13 | 452.3 | 2.9 (1.7–4.9) | **Ref** | | | |
| Married/cohabiting/in a relationship | 310 (28) | 7 | 330.8 | 2.1 (1.0–4.4) | 0.74 (0.29–1.85) | | | |
| Divorced/separated/widowed | 210 (19) | 4 | 117.2 | 3.4 (1.3–9.1) | 1.19 (0.39–3.64) | 0.710 | | |
| **Residence** | | | | | | | | |
| Non-fishing village | 928 (83) | 14 | 599.2 | 2.3 (1.4–3.9) | **Ref** | | **Ref** | |
| Fishing village | 187 (17) | 10 | 301.1 | 3.3 (1.8–6.2) | 1.42 (0.63–3.20) | 0.396 | 3.04 (1.05–8.78) | 0.040 |
| **Used a condom at last sex (past 3 months)** | | | | | | | | |
| No | 896 (80) | 16 | 695.3 | 2.3 (1.4–3.8) | **Ref** | | | |
| Yes | 219 (20) | 8 | 205.0 | 3.9 (2.0–7.8) | 1.70 (0.73–3.96) | 0.222 | | |
| **Had transactional sex (past 3 months)** | | | | | | | | |
| No | 129 (12) | 4 | 206.2 | 1.9 (0.7–5.2) | **Ref** | | | |
| Yes | 986 (88) | 20 | 694.1 | 2.9 (1.9–4.5) | 1.49 (0.51–4.35) | 0.470 | | |
| **Sex with a partner older ≥10 years (past 3 months)** | | | | | | | | |
| No | 415 (37) | 9 | 453.5 | 2.0 (1.0–3.8) | **Ref** | | | |
| Yes | 700 (63) | 15 | 446.8 | 3.4 (2.0–5.6) | 1.69 (0.74–3.87) | 0.213 | | |
| **Used recreational drugs (past 3 months)** | | | | | | | | |
| No | 905 (81) | 17 | 755.9 | 2.2 (1.4–3.6) | **Ref** | | **Ref** | |
| Yes | 210 (19) | 7 | 144.4 | 4.8 (2.3–10.2) | 2.16 (0.89–5.20) | 0.087 | 3.08 (1.19–7.99) | 0.021 |
| **Sex after consuming alcohol (past 3 months)** | | | | | | | | |
| No | 469 (42) | 10 | 390.3 | 2.6 (1.4–4.8) | Ref | | | |
| Yes | 646 (58) | 14 | 510.0 | 2.7 (1.6–4.6) | 1.07 (0.48 −2.41) | 0.868 | | |
| **Number of partners (past 3 months)** | | | | | | | | |
| ≤5 | 428 (38) | 11 | 529.6 | 2.1 (1.2–3.8) | **Ref** | | **Ref** | |
| ≥6 | 687 (62) | 13 | 370.7 | 3.5 (2.0–6.0) | 1.69 (0.76–3.77) | 0.201 | 1.96 (0.72–5.34) | 0.187 |
| **STI diagnosis/treatment (past 3 months)** | | | | | | | | |

*(Continued)*

| Baseline characteristics | Number of participants (%) | HIV incidence | | | Univariable analysis | | Multi-variable analysis | |
|---|---|---|---|---|---|---|---|---|
| | | N | PYO | IR/100PYO | IRR (95% CI) | P-value | aIRR (95% CI) | P-value |
| No | 681 (61) | 13 | 549.9 | 2.4 (1.4–4.1) | **Ref** | | | |
| Yes | 434 (39) | 11 | 350.4 | 3.1 (1.7–5.7) | 1.33 (0.59–2.96) | 0.489 | | |
| **Abnormal genital discharge (past 3 months)** | | | | | | | | |
| No | 532 (48) | 14 | 430.6 | 3.3 (1.9 −5.5) | **Ref** | | **Ref** | |
| Yes | 583 (52) | 10 | 469.7 | 2.1 (1.1–4.0) | 0.65 (0.29–1.47) | 0.306 | 0.38 (0.16–0.89) | 0.026 |
| **Genital ulcer (past 3 months)** | | | | | | | | |
| No | 818 (73) | 17 | 662.7 | 2.6 (1.6–4.1) | **Ref** | | | |
| Yes | 297 (27) | 7 | 237.6 | 2.9 (1.4–6.2) | 1.15 (0.48–2.77) | 0.758 | | |
| **Circumcised** | | | | | | | | |
| No (Male) | 65 (6) | 0 | 97.7 | 0.0 | | | | |
| Yes (Male) | 281 (25) | 5 | 354.9 | 1.4 (0.6–3.4) | | | | |
| Female | 769 (69) | 19 | 447.7 | 4.2 (2.7–6.7) | N/A | N/A | | |
| **Ever started PrEP** | | | | | | | | |
| No | 940 (84) | 17 | 612.1 | 2.8 (1.7-4.5) | **Ref** | | **Ref** | |
| Yes | 175 (16) | 7 | 288.2 | 2.4 (1.2-5.1) | 0.87 (0.36–2.11) | 0.765 | 0.66 (0.27–1.63) | 0.372 |
| **Calendar year**\*\* | | | | | | | | |
| 2018 | 188 (17) | 2 | 58.5 | 3.4 (0.9–13.7) | **Ref** | | | |
| 2019 | 365 (33) | 8 | 260.8 | 3.1 (1.5–6.1) | 0.90 (0.19–4.22) | | | |
| 2020 | 553 (50) | 9 | 345.2 | 2.6 (1.4–5.0) | 0.76 (0.16–3.53) | | | |
| 2021 | 757 (68) | 3 | 217.6 | 1.4 (0.4–4.3) | 0.40 (0.07–2.41) | | | |
| ≥2022 | 204 (18) | 2 | 18.2 | 11.0 (2.7–43.9) | 3.21 (0.45–22.77) | 0.337 | | |
| **Time in follow up**\*\* | | | | | | | | |
| First-year | 1115 (100) | 14 | 582.5 | 2.4 (1.4–4.1) | **Ref** | | | |
| Second year | 347 (31) | 8 | 250.3 | 3.2 (1.6–6.4) | 1.33 (0.56–3.17) | | | |
| Third year | 156 (14) | 2 | 67.5 | 3.0 (0.7–11.8) | 1.23 (0.28–5.42) | 0.808 | | |

n, number of HIV infections; PYO, person years of observation; IR, incidence rate; IRR, incidence rate ratio; aIRR, adjusted incidence rate ratio; §Occupations in the 'Other' category include: Professional/technical worker (64, 19%), sales/service worker (80, 24%), subsistence agricultural worker (55, 16%), craft and related trades worker (49, 15%), house helper/labourer (53, 16%), motorcyclist (26, 8%), unemployed (24, 7%), and others not listed (18, 15%). Participants could report more than one occupation, so the total responses exceed 337, which is the total number of study participants in this category (see S2 Table); N/A, not applicable; \*\*Participants could contribute PYO to multiple time periods, e.g., calendar years or follow-up periods.

Note: This analysis is based on data available as of Oct 2022, while data entry and cleaning were still in progress. Following the analysis cut-off date, two seroconversions initially reported in 2022 were redacted after confirmatory testing. Both were females, > 24 years of age, ≤ primary education, and resident in non-fishing communities. These updates had a minimal effect on the overall conclusions presented.

In adjusted analyses, incident HIV infection was significantly associated with female sex [adjusted incidence rate ratio (aIRR) =6.84 PYO, 95% CI 1.60–29.30], residence in a fishing village [aIRR=3.04 PYO, 95% CI 1.05–8.78], and use of recreational drugs in the past three months [aIRR=3.08 PYO, 95% CI 1.19–7.99]. Conversely, reported abnormal genital discharge in the past 3 months was significantly associated with a lower risk of incident HIV [aIRR=0.38 PYO, 95% CI 0.16–0.89] (Table 1).

## HIV incidence and risk factors during follow up

At the final follow-up assessment, incident HIV infection was significantly associated with reported engagement in sex after consuming alcohol in the past 3 months [aIRR=2.65 PYO, 95% CI 1.11–6.31] and with STI diagnosis or treatment during the same period [aIRR=2.52 PYO, 95% CI 1.09–5.80] (Table 2).

In the analyses accounting for repeated assessments of risk at individual level over time (S3 Table), reported engagement in sex after consuming alcohol in the past 3 months was significantly associated with HIV incidence [aIRR=4.16

**Table 2.  Risk factors for HIV acquisition among adults in Masaka, Uganda: Analysis of final follow-up data from an HIV vaccine preparedness study.**

| Characteristic at follow-up | Number of participants (%) | HIV incidence | | | Bi-variate analysis | | Multi-variable analysisΨ | |
|---|---|---|---|---|---|---|---|---|
| | | n | PYO | IR/100PYO | IRR (95% CI) | P-value | aIRR (95% CI) | P-value |
| **Overall** | 1115 (100) | 24 | 900.3 | 2.7 (1.8–4.0) | | | | |
| **Gender** | | | | | | | | |
| Male | 346 (31) | 5 | 452.6 | 1.1 (0.5–2.7) | **Ref** | | **Ref** | |
| Female | 769 (69) | 19 | 447.7 | 4.2 (2.7–6.7) | 3.84 (1.43–10.29) | 0.007 | 2.89 (0.74–11.33) | 0.127 |
| **Age at last HIV test** | | | | | | | | |
| ≤24 | 551 (49) | 13 | 379.7 | 3.4 (2.0–5.9) | 1.62 (0.73–3.62) | 0.239 | 1.61 (0.70–3.69) | 0.264 |
| >24 | 564 (51) | 11 | 520.6 | 2.1 (1.2–3.8) | **Ref** | | **Ref** | |
| **Occupation** | | | | | | | | |
| Other§ | 337 (30) | 3 | 366.8 | 0.8 (0.3–2.5) | **Ref** | | **Ref** | |
| Sex worker | 509 (46) | 7 | 209.0 | 3.3 (1.6–7.0) | 4.10 (1.06–15.84) | | 1.57 (0.34–7.30) | |
| Salon/Lodge/Bar worker | 181 (16) | 10 | 171.4 | 5.8 (3.1–10.8) | 7.13 (1.96–25.92) | | 3.46 (0.83–14.36) | |
| Subsistence fisheries worker | 88 (8) | 4 | 153.1 | 2.6 (1.0–7.0) | 3.19 (0.71–14.26) | 0.026 | 2.57 (0.48–13.86) | 0.225 |
| **Residence** | | | | | | | | |
| Non-fishing village | 928 (83) | 14 | 599.2 | 2.3 (1.4–3.9) | **Ref** | | **Ref** | |
| Fishing village | 187 (17) | 10 | 301.1 | 3.3 (1.8–6.2) | 1.42 (0.63–3.20) | 0.396 | 1.82 (0.65–5.10) | 0.254 |
| **Used a condom at last sex (past 3 months)** | | | | | | | | |
| No | 889 (80) | 19 | 695.3 | 2.7 (1.7–4.3) | **Ref** | | | |
| Yes | 226 (20) | 5 | 205.0 | 2.4 (1.0–5.9) | 0.89 (0.33–2.39) | 0.821 | | |
| **Had transactional sex (past 3 months)** | | | | | | | | |
| No | 251 (23) | 8 | 361.5 | 2.2 (1.1–4.4) | **Ref** | | | |
| Yes | 864 (77) | 16 | 538.8 | 3.0 (1.8–4.8) | 1.34 (0.57–3.13) | 0.497 | | |
| **Used recreational drugs (past 3 months)** | | | | | | | | |
| No | 944 (85) | 20 | 808.7 | 2.5 (1.6–3.8) | **Ref** | | | |
| Yes | 171 (15) | 4 | 91.6 | 4.4 (1.6–11.6) | 1.77 (0.60–5.16) | 0.300 | | |
| **Sex after consuming alcohol (past 3 months)** | | | | | | | | |
| No | 585 (52) | 8 | 540.5 | 1.5 (0.7–3.0) | **Ref** | | **Ref** | |
| Yes | 530 (48) | 16 | 359.8 | 4.4 (2.7–7.3) | 3.0 (1.3–7.0) | 0.011 | 2.65 (1.11–6.31) | 0.028 |
| **Number of partners (past 3 months)** | | | | | | | | |
| ≤5 | 499 (45) | 16 | 596.9 | 2.7 (1.6–4.4) | **Ref** | | | |
| ≥6 | 616 (55) | 8 | 303.4 | 2.6 (1.3–5.3) | 0.98 (0.42–2.30) | 0.969 | | |
| **STI diagnosis/treatment (past 3 months)** | | | | | | | | |
| No | 801 (72) | 14 | 721.8 | 1.9 (1.1–3.3) | **Ref** | | **Ref** | |
| Yes | 314 (28) | 10 | 178.5 | 5.6 (3.0–10.4) | 2.89 (1.28–6.50) | 0.010 | 2.52 (1.09–5.80) | 0.030 |
| **Abnormal genital discharge (past 3 months)** | | | | | | | | |
| No | 697 (63) | 14 | 660.3 | 2.1 (1.3–3.6) | Ref | | | |
| Yes | 418 (37) | 10 | 240.0 | 4.2 (2.2–7.7) | 1.96 (0.87–4.42) | 0.103 | | |
| **Genital ulcer (past 3 months)** | | | | | | | | |
| No | 934 (84) | 18 | 801.5 | 2.2 (1.4–3.6) | **Ref** | | | |
| Yes | 181 (16) | 6 | 98.8 | 6.1 (2.7–13.5) | 2.70 (1.07–6.81) | 0.035 | | |

n, number of HIV infections; PYO, person years of observation; IR, incidence rate; IRR, incidence rate ratio; aIRR, adjusted incidence rate ratio; ΨMulti-variable analyses were adjusted for gender, occupation, participants' residence, age at last HIV test and other predictors shown in the table.

PYO, 95% CI 1.60–10.85]. A weaker non-statistically significant association was observed for STI diagnosis or treatment in the past 3 months [aIRR = 2.06 PYO, 95% CI 0.84–5.06].

## Discussion

We observed a relatively high HIV incidence within this vaccine-preparedness cohort, indicating that HIV transmission continues to occur despite the availability of multiple prevention options. This ongoing transmission may partly reflect low uptake and suboptimal adherence to existing prevention strategies in some subpopulations [29,30]. Our findings are consistent with previous work in this population low uptake of PrEP, primarily due to pill burden, stigma, and the need for more time to make informed decisions about its use [31]. Although the observed incidence was high, it was lower than anticipated. This may reflect the increasing availability and uptake of a combination of effective interventions, including HCT, condom use, voluntary medical male circumcision, antiretroviral drugs for the prevention of mother-to-child transmission, PrEP, and treatment as prevention [32]. Previous studies in similar high-risk populations have reported HIV incidence rates ranging from 3.1 to 9.3 per 100PYs [33]. As noted above, declining HIV incidence in high-risk populations may present challenges for the conduct of HIV prevention trials, as lower incidence reduces the ability to measure the effectiveness of new interventions.

In our study, HIV incidence was significantly higher among women than among men. Women are at an increased risk of HIV acquisition due to a combination of biological, behavioral, socioeconomic, cultural and structural factors [34]. Structural barriers, such as limited access to HIV prevention services, lower education opportunities, limited skill development options, intimate partner violence, and migration driven by economic hardship, may further exacerbate their vulnerability to HIV infection [35]. Younger women, in particular, have become a major focus of HIV prevention interventions efforts [36].

We found that individuals residing in fishing communities were at higher risk of acquiring HIV than their counterparts in other communities. People living in fishing communities in sub-Saharan Africa are known to face increased HIV risk due to several factors, including high mobility, engagement in transactional sex, multiple sexual partnerships, heavy alcohol use, inadequate health infrastructure, and limited access to healthcare services [33,37,38]. Additionally, newer interventions such as oral PrEP remain underutilized in these communities, which may hinder efforts to reduce HIV incidence [39].

Illicit drug use can directly or indirectly increase the likelihood of engaging in risky sexual behaviors, thereby increasing the risk of HIV acquisition. This may explain our finding that use of reactional drugs and alcohol was associated with higher HIV incidence. Recreational drug use is relatively common among high-risk groups, including sex workers and individuals in fishing communities in SSA [40]. Both recreational drug use and alcohol consumption have been associated with increased likelihood of sexual encounters, inconsistent condom use, and other high-risk sexual practices [41].

STIs are known risk factors for HIV acquisition. They can cause genital inflammation and ulceration, which facilitate viral entry, particularly among women [42]. We found that both a diagnosis or treatment of an STI and a history of genital ulcers were significantly associated with HIV acquisition. Interestingly, abnormal genital discharge was associated with a reduced risk of HIV acquisition. This unexpected finding may reflect unmeasured confounding or possible misclassification. For example, some female participants may have misreported normal vaginal discharge associated with menstruation or the use of oestrogen-containing contraception as abnormal [43]. Alternatively, the reported discharge may have resulted from disturbances of the normal vaginal flora rather than from an STI. Although vaginal discharge itself does not directly protect against HIV, a healthy vaginal microbiome, dominated by *Lactobacillus species*, plays a key role in reducing HIV susceptibility by maintaining an acidic pH, strengthening the mucosal barrier, and preventing inflammation [44]. Nevertheless, other studies have similarly reported no clear association between abnormal vaginal discharge and an increased risk of HIV acquisition [45].

A major limitation of this study is that 22% of individuals did not attend any follow-up visits. These individuals were more likely to be young (<25years), female, recreational drug users, and persons who engage in transactional sex [46], subpopulations that are also associated with a higher risk of HIV acquisition. Their absence from follow-up likely introduced selection bias, potentially leading to an underestimation of HIV incidence. LTFU was also higher during the COVID-19 restriction

period (2020–2021) compared with the pre-pandemic period (2018–2019). To support retention, several strategies were implemented, including collection of detailed locator data, short clinic visits (1–2 hours), flexible reimbursement of transport costs, immediate follow-up of missed visits (≤7 days) via phone and/or home visits, and community engagement meetings.

Additionally, data on HIV risk factors were self-reported. Despite being collected by well-trained and experienced staff, these may have been affected by recall bias, social desirability bias, or misreporting. To mitigate these biases, multiple questions were used to verify and crosscheck responses. The inclusion of objective measures, such as laboratory testing for STI testing, could have provided biological confirmation of HIV risk but was not feasible in this study.

HIV infection was diagnosed using antibody-based rapid diagnostic tests, which may not identify very recent infections, before seroconversion has occurred, potentially resulting in delayed diagnosis and misclassification of HIV status. Quarterly HIV testing substantially mitigated this risk by increasing the likelihood of detecting new HIV infections during follow-up, although some underestimation of HIV incidence may still have occurred.

## Conclusions

HIV incidence remains high, especially among women and residents of fishing communities, despite the availability of effective prevention interventions such as oral PrEP. To address this issue, it is crucial to strengthen HIV prevention and care services specifically tailored to these vulnerable populations. Targeted HIV education and community outreach programs could enhance access to oral PrEP in fishing communities by addressing stigma, low awareness, and logistical barriers to service uptake. Additionally, counselling and support groups can play a crucial role in mitigating risk associated with substance use and encouraging PrEP uptake and adherence. Prevention efforts should prioritise targeted interventions that focus on reducing high-risk behaviours, including recreational drug and alcohol use, while also strengthening the prevention, screening, and treatment of STIs.

As the use of placebo-controlled trials becomes less feasible due to the increasing availability of highly effective HIV prevention tools, future HIV prevention trials must explore innovative methodologies to assess the efficacy of new interventions. Some of the proposed approaches include the use of registration cohorts, such as the one described in this paper, to estimate HIV incidence in real-world settings without the experimental intervention; recency assays to determine recent HIV infection at trial screening; external placebo arms using incidence data from previous placebo-controlled HIV prevention trials as a reference; biological markers of HIV incidence to assess HIV exposure and infer intervention efficacy; drug concentration measurements to evaluate pharmacokinetic effectiveness of prevention products; and immune biomarkers, such as antibody responses, to help predict the protective effectiveness of emerging HIV prevention strategies [47].

## Supporting information

**S1 Fig. Serial HIV testing algorithm for persons above 18 months of age.**
(DOCX)

**S1 Table. Comparison of baseline characteristics between participants who did not attend any follow-up visit and those who did in a HIV vaccine preparedness study at Masaka, Uganda.**
(DOCX)

**S2 Table. HIV incidence by participant occupation among participants classified in those categorised the "Other" occupational category in** Table 1**.**
(DOCX)

**S3 Table. Risk indicators of HIV acquisition among adults in a HIV vaccine preparedness study, Masaka, Uganda, accounting for repeated risk assessments over time.**
(DOCX)

## Acknowledgments

We thank all the participants and staff of the PrEPVacc registration cohort study at the MRC/UVRI and LSHTM Uganda Research Unit Masaka Clinical Research Centre.

This study is also made possible by the support of the American People through the U.S. President's Emergency Plan for AIDS Relief (PEPFAR) through United States Agency for International Development (USAID). The contents of this publication are the sole responsibility of the PrEPVacc Team and do not necessarily reflect the views of PEPFAR, USAID, or the United States Government.

## Author contributions

**Conceptualization:** Sylvia Kusemererwa, Sheila Kansiime, Gertrude Mutonyi, Ayoub Kakande, Julie Fox, Sheena McCormack, Pontiano Kaleebu, Eugene Ruzagira.

**Data curation:** Sylvia Kusemererwa, Sheila Kansiime, Gertrude Mutonyi, Ayoub Kakande, Shamim Nabukenya, Aeron Namirembe, Naphtal Erima, Julie Fox, Sheena McCormack, Pontiano Kaleebu, Eugene Ruzagira.

**Formal analysis:** Sheila Kansiime.

**Funding acquisition:** Sheena McCormack, Pontiano Kaleebu, Eugene Ruzagira.

**Investigation:** Sylvia Kusemererwa, Sheila Kansiime, Shamim Nabukenya, Doreen Joan Asio, Elizabeth Mbabazi Atuhura, Shamim Ssendagire, Martin Onyango, Julie Fox, Sheena McCormack, Pontiano Kaleebu, Eugene Ruzagira.

**Methodology:** Sylvia Kusemererwa, Sheila Kansiime, Gertrude Mutonyi, Ayoub Kakande, Shamim Nabukenya, Aeron Namirembe, Naphtal Erima, Doreen Joan Asio, Elizabeth Mbabazi Atuhura, Shamim Ssendagire, Martin Onyango, Julie Fox, Sheena McCormack, Pontiano Kaleebu, Eugene Ruzagira.

**Project administration:** Sylvia Kusemererwa, Sheena McCormack, Pontiano Kaleebu, Eugene Ruzagira.

**Resources:** Sheena McCormack, Pontiano Kaleebu, Eugene Ruzagira.

**Software:** Sheila Kansiime, Gertrude Mutonyi, Shamim Nabukenya, Aeron Namirembe, Naphtal Erima.

**Supervision:** Sylvia Kusemererwa, Ayoub Kakande, Shamim Ssendagire, Martin Onyango, Julie Fox, Sheena McCormack, Pontiano Kaleebu, Eugene Ruzagira.

**Validation:** Sylvia Kusemererwa, Sheila Kansiime, Gertrude Mutonyi, Ayoub Kakande, Shamim Nabukenya, Julie Fox, Sheena McCormack, Pontiano Kaleebu, Eugene Ruzagira.

**Visualization:** Sylvia Kusemererwa, Sheila Kansiime, Ayoub Kakande, Shamim Nabukenya, Julie Fox, Sheena McCormack, Pontiano Kaleebu, Eugene Ruzagira.

**Writing – original draft:** Sylvia Kusemererwa, Sheila Kansiime, Eugene Ruzagira.

**Writing – review & editing:** Sylvia Kusemererwa, Sheila Kansiime, Gertrude Mutonyi, Ayoub Kakande, Shamim Nabukenya, Aeron Namirembe, Naphtal Erima, Doreen Joan Asio, Elizabeth Mbabazi Atuhura, Shamim Ssendagire, Martin Onyango, Julie Fox, Sheena McCormack, Pontiano Kaleebu, Eugene Ruzagira.

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
