## [Decision Letter · Decision Letter 0]

23 Jan 2025

PONE-D-24-44704HIV incidence and associated risk factors among adults enrolled in an HIV vaccine preparedness cohort in Masaka, UgandaPLOS ONE

Dear Dr. Kusemererwa,

Thank you for submitting your manuscript to PLOS ONE. After careful consideration, we feel that it has merit but does not fully meet PLOS ONE’s publication criteria as it currently stands. Therefore, we invite you to submit a revised version of the manuscript that addresses the points raised during the review process.

We look forward to receiving your revised manuscript.

Kind regards,

Fred Nuwaha

Academic Editor

PLOS ONE

Journal Requirements:

“The PrEPVacc registration cohort study was funded by The second European & Developing Countries Clinical Trials Partnership (EDCTP2) (Grant number: RIA-2016V-1644).”

Reviewers' comments:

Reviewer's Responses to Questions

**Comments to the Author**

1. Is the manuscript technically sound, and do the data support the conclusions?

Reviewer #1: Partly

Reviewer #2: Yes

2. Has the statistical analysis been performed appropriately and rigorously?

Reviewer #1: No

Reviewer #2: Yes

3. Have the authors made all data underlying the findings in their manuscript fully available?

Reviewer #1: Yes

Reviewer #2: Yes

4. Is the manuscript presented in an intelligible fashion and written in standard English?

Reviewer #1: Yes

Reviewer #2: Yes

5. Review Comments to the Author

Reviewer #1: Reviewing of the paper titled

HIV incidence and associated risk factors among adults enrolled in an HIV vaccine preparedness cohort in Masaka, Uganda

Thank you for the opportunity to review this article, which has the potential to make a significant contribution to HIV research, particularly in the context of the "First 95" initiative. The article presents a study conducted as part of the PrEPVacc registration cohort study, aimed at estimating HIV incidence and identifying associated risk factors within a high-risk population in preparation for the PrEPVacc HIV vaccine trial. Conducted between 2018 and 2022 across multiple clinical research sites in East Africa, the study focuses particularly on Masaka, Uganda.

While the study offers valuable insights into HIV incidence and risk factors, there are several major flows that need to be addressed. These concerns primarily relate to the identification of recent HIV infections, study methodology, and the interpretation of certain results.

I set out my concerns in greater detail below.

Major comments

Introduction

The introduction begins with a compelling statistic, underlining the urgency of HIV prevention efforts, particularly in regions like Eastern and Southern Africa, which account for a significant proportion of new infections. The discussion of key populations and their disproportionate burden of HIV provides a well-justified rationale for targeting high-risk groups in research. However, there are areas where the introduction could be further refined to strengthen its impact and clarity within an epidemiological context.

Lines 67 – 71: Although the authors mention that current HIV incidence data are limited, they do not specify the scope or nature of these gaps. For instance, it would strengthen the introduction if the authors detailed regional disparities or methodological challenges in existing incidence data, particularly in populations similar to those studied.

Lines 67 – 79: In the introduction of this manuscript, the extensive discussion on the challenges associated with implementing clinical trials for prevention interventions, such as PrEP or HIV vaccines, appears unnecessary. It would be more appropriate for the authors to condense this section to maintain focus and conciseness.

Moreover, authors should briefly outline methodological innovations that address these challenges. This would provide a forward-looking perspective and demonstrate the study’s contribution to advancing HIV prevention research.

Lines 80 – 85 : Authors should :

• Clearly outline what sets this study apart from previous research.

• Justify the study's geographic focus with specific local epidemiological data.

• Include a concise literature review on HIV incidence and risk factors in Masaka to underscore the study's relevance and originality.

Methods and Material

Line 141 – 143 : How were participants in seroconversion identified? The use of the national screening algorithm based on rapid tests may not be sufficient, as it could yield false negatives. Therefore, it is necessary to conduct more advanced laboratory testing, such as recent infection tests (within a specific timeframe, e.g., 12 months), to confirm that the cases reported during follow-up are indeed new infections and not pre-existing ones.

Lines 145 – 147: This approach of analysis does not seem ideal given the study's cohort design. The major limitation of this approach is that it considers only inter-individual variability while completely ignoring intra-individual variability across all factors influencing the occurrence of new HIV infections. This effectively reduces the cohort study design to two cross-sectional studies, which is not appropriate.

We therefore recommend that the authors, instead of using two univariable and multivariable Poisson regression models, consider using a Poisson model with random effects or a marginal Poisson model with a Generalized Estimating Equations (GEE) approach. The working correlation matrix should be exchangeable, determined by the information criterion (QICu). Since the authors use STATA 18.0, we recommend that they review the documentation and utilize the xtpoisson command, which supports the implementation of these models.

Some critical subsections appear to be missing in the methodology:

• A detailed description of the study site.

• The sampling method employed within the community.

• The calculation of the sample size required to measure incidence in this population.

• A more precise description of the PrePVacc trial, including the primary and secondary endpoints.

Results

Lines 173 – 178: A substantial proportion of participants (22%) did not attend any follow-up visits, and only 38% completed the study. These figures raise concerns about potential attrition bias. The reasons for dropout are mentioned but not explored in depth. Authors should Perform a sensitivity analysis to assess the impact of attrition on the results and provide a detailed discussion of measures taken to minimize dropout.

Lines 224 : Table 1: Concerns Regarding the Categorization of the Variable "Occupation".

I have significant concerns about the categorization of the variable Occupation. The "Other" category encompasses 30% of the enrolled population (330 participants), which is an unusually high proportion. Additionally, this category includes diverse groups such as professional/technical workers, sales/service workers, office clerks, and students, which introduces considerable heterogeneity. The behaviors and HIV infection risks of students, for instance, are likely to differ significantly from those of professional or technical workers. Aggregating these distinct subgroups into a single category is methodologically inappropriate.

The use of this aggregated category in the analysis—and particularly in the interpretation of other groups, as observed on line 211—raises concerns about the validity and specificity of the findings.

We strongly recommend that the authors disaggregate this variable into more meaningful and homogeneous subcategories and reconsider its inclusion in the analyses to ensure more accurate and interpretable results. Using this aggregated category in the analysis, and particularly in the interpretation of other groups (as done on line 211), is therefore problematic.

We strongly recommend that the authors refine this variable by disaggregating its components and reconsider its inclusion in the analyses.

Line 246 : Table 2: The authors have not considered certain variables from the analysis in Table 1, such as marital status or occupation, which could have changed over the 4-year follow-up period and, consequently, influenced the risk of HIV infection. If the authors assessed these factors at each visit, we recommend treating these variables as time-dependent in the random effects Poisson model or the marginal Poisson model, as we suggested earlier.

Discussion

Line 253: The authors mention low PrEP uptake in the study population as a potential reason for persistent HIV transmission. While this is an important observation, it would be valuable to delve deeper into why PrEP uptake is low. Are there barriers to access (e.g., cost, stigma, lack of awareness), or are there other socio-behavioral factors? This could inform targeted interventions to improve PrEP coverage and adherence.

Line 260 – 265: The discussion largely focuses on descriptive findings but could benefit from more in-depth exploration of the theoretical mechanisms underlying the observed associations. For example, the authors could expand on how specific socio-economic and cultural factors (e.g., gender inequality, lack of access to healthcare) might exacerbate HIV transmission, particularly in subpopulations like women and residents of fishing communities.

Lines 292 – 294 : The authors briefly mention the limitations of the data, such as selection bias due to participant dropout and the potential for underreporting of HIV risk factors. However, the statistical methods used to handle these issues are not discussed in detail. It would be beneficial for the authors to address whether any sensitivity analyses were performed to account for potential biases or missing data, or if any imputation techniques were used.

Conclusion

Line 301 – 303 : The discussion on prevention strategies is relevant, but it could be enhanced by offering more concrete recommendations for public health interventions. For example, given the identified risk factors (e.g., low PrEP uptake, alcohol and drug use), the authors could discuss specific interventions that could address these issues within the study population or similar high-risk groups.

Line 306: While the authors suggest exploring innovative methodologies for future trials, they could clarify what types of methodologies they consider to be the most promising. For example, what new approaches could be explored to assess the efficacy of interventions, especially in populations that have traditionally been harder to reach? Specific examples of innovative methodologies (e.g., digital health tools, community-based interventions) would make this recommendation more actionable.

Minor comments

Title:

I suggest the title: “Estimating HIV incidence and assessing associated risk factors among adults: Evidence from the 2018 HIV vaccine preparedness cohort in Masaka, Uganda”

Introduction

Lines 58-59 : Integrate updated HIV epidemiological statistics to reflect the most recent trends (2023) and provide an overview of the evolution over the past three decades. The authors are encouraged to reference THE 2024 GLOBAL AIDS UPDATE report.

Lines 67 – 69 : Authors should support this statement with evidence (figures).

Methods and Material

Lines 103 – 106 : The reliance on HIV counseling and testing (HCT) outreach programs for recruitment may introduce selection bias by disproportionately including individuals already engaged with healthcare services. This could limit the generalizability of the findings to the broader at-risk population. Authors should include a discussion of the potential biases introduced by the recruitment strategy and consider alternative methods for reaching less-engaged populations.

Lines 107-108 : The description of screening and enrollment procedures is concise but lacks detail on how potential participants were approached and convinced to join the study.

Lines 121 – 126: The sentence is too long. It would be preferable for the authors to present the complete participation criteria of the study using bullet points.

Line 129: The authors should also provide evidence or studies on the intrinsic performance metrics of the screening algorithm in their country, including sensitivity, specificity, and, most importantly, the positive and negative predictive values, to reassure readers about the reliability of the results.

Lines 129 – 134: The description of laboratory procedures, while adequate, could benefit from more detail on quality control measures for HIV testing and pregnancy screening to ensure reproducibility. Authors should Provide detailed information on quality assurance practices, including, where applicable, external quality control and validation or inter-laboratory comparisons.

Line 134: For enhanced clarity, the authors should present the HIV screening and testing algorithm using a diagram.

Lines 135 – 136: What is the outcome for pregnant women in the study? The authors should clarify the follow-up procedures specifically for pregnant women!

Lines 150 – 154 : The methods for collecting data on HIV risk behaviors rely on self-reports, which are susceptible to memory bias and social desirability bias. Discuss these limitations explicitly and, if available, include validation measures (e.g., biomarkers or triangulation) to corroborate self-reported data.

Line 159 – 160 : The authors should justify why they decided to include age and gender a priori in the multivariable model.

Results

Lines 178 -181 : The authors should provide a table or figure to effectively support these results.

Line 202 : Fig 1 : The authors should provide a more explicit explanation for the exclusion reason labeled as "investigator discretion" and include it under the "Other" category.

Line 199 Fig 1 : Here, the authors present 370 participants who completed the cohort study and will be enrolled in the PrePVacc trial, along with 30 participants who completed the study but will not be enrolled. Could the authors clarify the distinction between this cohort study and the PrePVacc trial? If the two studies are independent, why is the PrePVacc trial mentioned in this context?

Line 208 : If the 24 new infections were identified using rapid HIV diagnostic tests, this is insufficient to draw definitive conclusions, as these tests do not confirm whether the infections are indeed new. Additional recent HIV infection assays should be conducted. If the authors are unable to perform such tests, they must acknowledge this limitation in the discussion.

Line 209 – 221 : Here, the authors interpret the results of the model based on baseline factors. However, the interpretation includes both univariable and multivariable analyses, leading to redundancy (e.g., for the variable "Sex"). Generally interpretation focuses on the results of the multivariable analysis, as it accounts for adjustments across other factors, thereby providing a more accurate representation of the phenomenon in the study population, where these factors operate simultaneously.

Lines 224 : Table 1: Could the authors explain the "N/A" for the variable "Circumcised" in both the univariable and multivariable analyses?

Discussion

Line 256 : While the authors present the high incidence of HIV and its association with certain risk factors, the discussion could benefit from a more nuanced exploration of why the incidence rate was "lower than expected" (line 256). Moreover, the authors should specified the expected incidence rate.

Lines 282 – 291: The authors note that abnormal genital discharge was unexpectedly associated with a reduced risk of HIV acquisition. While they mention potential misclassification or confounding factors, it would be helpful to further elaborate on how other unmeasured variables (e.g., hormonal fluctuations, non-STI-related vaginal infections) might explain this result. A more detailed exploration of potential confounders would enhance the robustness of the interpretation.

Lines 295 – 298: The authors mention that HIV risk factors were self-reported, which is a common limitation in cohort studies. While this is acknowledged, the authors could strengthen the discussion by elaborating on how these biases (e.g., recall bias, social desirability bias) could specifically affect their findings. Furthermore, it would be useful to consider how the inclusion of biomarkers or objective measures of risk (e.g., STI testing) could have enhanced the reliability of their data.

Conclusion

Lines 303 – 305: The authors could further explore how the findings of this study could influence public health policies and programs. For example, how might public health systems prioritize HIV prevention in high-risk populations? Expanding on this point would provide a clearer path from research findings to practical, policy-oriented action.

Reviewer #2: This is an important paper from a study to determine HIV incidence in the post-PrEP scale-up era. The study is quite useful for planning HIV prevention studies, including vaccines. However, the study design and results need some clarification.

1. The authors state that “Additionally, time-related predictors, including duration of follow-up and the calendar period which person-years were accrued”. This statement is unclear as the variables that were time-updated are not mentioned. Specifically, since sex behaviours (and other behaviours) change during the study these variables (eg sex with multiple partners, alcohol, STIs etc) could have been modelled as time-updated variables

2. Please clarify whether all seroconverted patients were linked to care and started on antiretroviral therapy

3. Was PrEP made available to all eligible participants. By the Ugandan guidelines, all the participants are eligible for PrEP. It is also not clear whether these patients received basic HIV prevention health education. These issues should be discussed in the ethics section

4. The use of “time-related variables” is confusing as it will be confused with time-updated variables. Please use words such as cohort year or number of follow-up years.

5. The number of incident cases was few and a brief table with their characteristics (Age, sex, and follow-up time) would be helpful, especially as a supplementary table

6. The other limitation is that there was high LTFU and the number of events is few. As can be seen, the Confidence intervals are quite wide.

7. If this study is to plan for candidates for HIV vaccine study, then it would be nice to regress the predictors of those retained in care as these are possible participants with a high risk of HIV

6. PLOS authors have the option to publish the peer review history of their article (what does this mean?). If published, this will include your full peer review and any attached files.

Reviewer #1: **Yes:** Cyprien KENGNE-NDE

Reviewer #2: **Yes:** Dathan M Byonanebye

---

## [Author Response · Author response to Decision Letter 1]

16 Apr 2025

Response sheet for submission ID PONE-D-24-44704 R1: HIV incidence and associated risk factors among adults enrolled in an HIV vaccine preparedness cohort in Masaka, Uganda

Comments and responses

We wish to thank the reviewers for their valuable comments. Please find below a point-by-point response to comments raised.

Reviewer #1

Comment 1

Thank you for the opportunity to review this article, which has the potential to make a significant contribution to HIV research, particularly in the context of the "First 95" initiative. The article presents a study conducted as part of the PrEPVacc registration cohort study, aimed at estimating HIV incidence and identifying associated risk factors within a high-risk population in preparation for the PrEPVacc HIV vaccine trial. Conducted between 2018 and 2022 across multiple clinical research sites in East Africa, the study focuses particularly on Masaka, Uganda.

While the study offers valuable insights into HIV incidence and risk factors, there are several major flows that need to be addressed. These concerns primarily relate to the identification of recent HIV infections, study methodology, and the interpretation of certain results. I set out my concerns in greater detail below.

Major comments

Comment 2: The introduction begins with a compelling statistic, underlining the urgency of HIV prevention efforts, particularly in regions like Eastern and Southern Africa, which account for a significant proportion of new infections. The discussion of key populations and their disproportionate burden of HIV provides a well-justified rationale for targeting high-risk groups in research. However, there are areas where the introduction could be further refined to strengthen its impact and clarity within an epidemiological context.

Lines 67 – 71: Although the authors mention that current HIV incidence data are limited, they do not specify the scope or nature of these gaps. For instance, it would strengthen the introduction if the authors detailed regional disparities or methodological challenges in existing incidence data, particularly in populations similar to those studied.

Response: We appreciate the reviewer for their suggestion. We have revised the introduction to highlight the disparities in HIV incidence within and between different key populations in sub-Saharan Africa.

Comment 3: Lines 67 – 79: In the introduction of this manuscript, the extensive discussion on the challenges associated with implementing clinical trials for prevention interventions, such as PrEP or HIV vaccines, appears unnecessary. It would be more appropriate for the authors to condense this section to maintain focus and conciseness.

Moreover, authors should briefly outline methodological innovations that address these challenges. This would provide a forward-looking perspective and demonstrate the study’s contribution to advancing HIV prevention research.

Response: We thank the reviewer for this insight. We have included details on methodological innovations and recommendations that address challenges around estimating HIV incidence in HIV prevention trials, namely: adjusting for pre-existing infections, product adherence, and discontinuation in sample size calculations, while also developing biomarkers and alternative trial designs.

Comment 4: Lines 80 – 85: Authors should:

• Clearly outline what sets this study apart from previous research.

• Justify the study's geographic focus with specific local epidemiological data.

• Include a concise literature review on HIV incidence and risk factors in Masaka to underscore the study's relevance and originality.

Response: We thank the reviewer for their suggestions. The introduction has been revised accordingly.

Methods and Material

Comment 5: Line 141 – 143: How were participants in seroconversion identified? The use of the national screening algorithm based on rapid tests may not be sufficient, as it could yield false negatives. Therefore, it is necessary to conduct more advanced laboratory testing, such as recent infection tests (within a specific timeframe, e.g., 12 months), to confirm that the cases reported during follow-up are indeed new infections and not pre-existing ones.

Response: As outlined in the manuscript's introduction, the primary aim of the PrEPVacc registration cohort was to estimate HIV incidence in populations targeted for the PrEPVacc trial. Only individuals who tested HIV-negative at baseline and met the eligibility criteria were enrolled. Those who tested HIV-positive were ineligible and were referred for HIV care. Recency testing was not performed, as the study did not aim to distinguish newly acquired HIV infections from prevalent cases.

HIV testing was at screening, enrollment, and at quarterly follow-up visits was performed in accordance with Uganda’s national guidelines, using the Serial HIV Testing Algorithm for individuals above 18 months of age. This algorithm has a sensitivity of 99.2%, specificity of 99.1%, positive predictive value of 99.0% and negative predictive value of 99.2% (Kaleebu, Kitandwe et al. 2018).

To maintain quality control, study staff underwent training and certification by a national provider to conduct HIV testing services. Internal quality control was performed before testing participant samples, and the study site participated in Uganda’s national external quality assurance (EQA) program, with EQA conducted quarterly. A section detailing these quality control procedures has been added to the revised manuscript under laboratory procedures.

Comment 6: Lines 145 – 147: This approach of analysis does not seem ideal given the study's cohort design. The major limitation of this approach is that it considers only inter-individual variability while completely ignoring intra-individual variability across all factors influencing the occurrence of new HIV infections. This effectively reduces the cohort study design to two cross-sectional studies, which is not appropriate.

We therefore recommend that the authors, instead of using two univariable and multivariable Poisson regression models, consider using a Poisson model with random effects or a marginal Poisson model with a Generalized Estimating Equations (GEE) approach. The working correlation matrix should be exchangeable, determined by the information criterion (QICu). Since the authors use STATA 18.0, we recommend that they review the documentation and utilize the xtpoisson command, which supports the implementation of these models.

Some critical subsections appear to be missing in the methodology:

• A detailed description of the study site.

• The sampling method employed within the community.

• The calculation of the sample size required to measure incidence in this population.

• A more precise description of the PrePVacc trial, including the primary and secondary endpoints.

Response: We appreciate the reviewer’s detailed comment. The analytical approach used in this paper is a standard method for analyzing incidence data from a single study site. Similar approaches have been applied in other studies (Ahmed, Lutalo et al. 2001, De Schacht, Mabunda et al. 2014, Westreich, Jamal et al. 2014, Gómez-Olivé, Houle et al. 2020).

If this were a multi-site study or one where the same individual was assessed multiple times for an outcome (e.g., repeated blood pressure measurements), incorporating random effects or using a GEE model would be appropriate. However, in our case, each participant had a single outcome (HIV status at the end of follow-up), and the study was conducted at a single site (Masaka). Therefore, there was no intra-individual variability or clear random effect to account for.

We have also added separate subsections in the methodology section for study design, study site, and population, outlining the sample selection process. Additionally, we have provided a more detailed description of the PrEPVacc trial in the revised manuscript.

For the sample size, a post hoc sample size calculation showed ≥900 person-years of follow-up (PYO), and an incidence rate of ~2.7/100PYO provided a confidence interval with a precision ≤3/ 100PYO. Additionally, with 900 person years of follow up, we had ≥80% power to identify factors with an incident rate ratio ≥2.5.

Results

Comment 7: Lines 173 – 178: A substantial proportion of participants (22%) did not attend any follow-up visits, and only 38% completed the study. These figures raise concerns about potential attrition bias. The reasons for dropout are mentioned but not explored in depth. Authors should perform a sensitivity analysis to assess the impact of attrition on the results and provide a detailed discussion of measures taken to minimize dropout.

Response: Thank you for this comment. We acknowledge that the considerable attrition in this study may have introduced bias in the observed incidence rate, and we have explicitly addressed this limitation in the manuscript. A separate analysis on LTFU in this cohort (Kabarambi, Kansiime et al. 2022) found that individuals at potentially higher risk of HIV—including young people (<25 years), females, recreational drug users, and those engaged in transactional sex—were more likely to be lost to follow-up. This suggests that our HIV incidence estimate may be an underestimate.

Additionally, participant follow-up was significantly impacted by disruptions caused by COVID-19. However, we do not consider a sensitivity analysis necessary, as it would require hypothetical assumptions about incidence among those lost to follow-up. Instead, we have clearly acknowledged this limitation and detailed the measures taken in the study to minimize dropout.

Comment 8: Lines 224: Table 1: Concerns Regarding the Categorization of the Variable "Occupation".

I have significant concerns about the categorization of the variable Occupation. The "Other" category encompasses 30% of the enrolled population (330 participants), which is an unusually high proportion. Additionally, this category includes diverse groups such as professional/technical workers, sales/service workers, office clerks, and students, which introduces considerable heterogeneity. The behaviors and HIV infection risks of students, for instance, are likely to differ significantly from those of professional or technical workers. Aggregating these distinct subgroups into a single category is methodologically inappropriate.

The use of this aggregated category in the analysis—and particularly in the interpretation of other groups, as observed on line 211—raises concerns about the validity and specificity of the findings.

We strongly recommend that the authors disaggregate this variable into more meaningful and homogeneous subcategories and reconsider its inclusion in the analyses to ensure more accurate and interpretable results. Using this aggregated category in the analysis, and particularly in the interpretation of other groups (as done on line 211), is therefore problematic.

We strongly recommend that the authors refine this variable by disaggregating its components and reconsider its inclusion in the analyses.

Response: Thank you for this comment. We appreciate the rationale behind your suggestion. However, the categorization of the occupation variable was based on the known HIV epidemiology in our study setting. Sex workers, salon/bar/lodge workers, and fisher-folk have consistently been identified as populations at exceptionally high risk of HIV acquisition. In contrast, while other occupations, encompass diverse groups (e.g., house helps vs. professional/technical workers); they have not been specifically highlighted as high-risk groups in the same way. Indeed, in our analysis, this group has the lowest HIV incidence. In the revised manuscript, we provide a breakdown of the occupations in this group in a footer under Table 1.

Comment 9: Line 246: Table 2: The authors have not considered certain variables from the analysis in Table 1, such as marital status or occupation, which could have changed over the 4-year follow-up period and, consequently, influenced the risk of HIV infection. If the authors assessed these factors at each visit, we recommend treating these variables as time-dependent in the random effects Poisson model or the marginal Poisson model, as we suggested earlier.

Response: We agree with the reviewer on this comment; however, these data were only collected at baseline. As such, we are unable to account for changes during follow-up

Discussion

Comment 10: Line 253: The authors mention low PrEP uptake in the study population as a potential reason for persistent HIV transmission. While this is an important observation, it would be valuable to delve deeper into why PrEP uptake is low. Are there barriers to access (e.g., cost, stigma, lack of awareness), or are there other socio-behavioral factors? This could inform targeted interventions to improve PrEP coverage and adherence.

Response: We thank the reviewer for their observation and concur with their view. The reasons for low uptake have been studied and published using data from the same cohort (Kusemererwa, Kansiime et al. 2021). Briefly, uptake was low in this population mainly because of fear of pill burden, stigma, and participants needing more time to decide to start oral PrEP. We have included a sentence in the revised manuscript summarizing the key reasons for non-uptake.

Comment 11: Line 260 – 265: The discussion largely focuses on descriptive findings but could benefit from more in-depth exploration of the theoretical mechanisms underlying the observed associations. For example, the authors could expand on how specific socio-economic and cultural factors (e.g., gender inequality, lack of access to healthcare) might exacerbate HIV transmission, particularly in subpopulations like women and residents of fishing communities.

Response: We appreciate the reviewer for their suggestion. In the revised manuscript, we have expanded our discussion on the of gender dynamics and access to healthcare in shaping HIV risk. Additionally, we had already provided some detail on the factors influencing HIV risk among residents of fishing communities in paragraph 3.

Comment 12: Lines 292 – 294: The authors briefly mention the limitations of the data, such as selection bias due to participant dropout and the potential for underreporting of HIV risk factors. However, the statistical methods used to handle these issues are not discussed in detail. It would be beneficial for the authors to address whether any sensitivity analyses were performed to account for potential biases or missing data, or if any imputation techniques were used.

Response: We thank the reviewer for this comment. As noted in our response to comment 6, we have acknowledged the limitations associated with high attrition and have referenced a published paper on loss to follow up in our cohort (Kabarambi, Kansiime et al. 2022). We have also explained why we did not perform sensitivity analyses.

Conclusion

Comment 13: Line 301 – 303: The discussion on prevention strategies is relevant, but it could be enhanced by offering more concrete recommendations for public health interventions. For example, given the identified risk factors (e.g., low PrEP uptake, alcohol and drug use), the authors could discuss specific interventions that could address these issues within the study population or similar high-risk groups.

Response: We thank the reviewer for their suggestion. In the revised manuscript, we recommend targeted HIV education and outreach programs as way of enhancing access to oral PrEP in fishing communities through addressing stigma and logistical barriers. Additionally, counselling and support groups can play a crucial role in mitigating substance use-related risks and encouraging PrEP uptake.

Comment 14: Line 306: While the authors suggest exploring innovative methodologies for future trials, they could clarify what types of methodologies they consider to be the most promising. For example, what new approaches could be explored to assess the efficacy of interventions, especially in populations that have traditionally been harder to reach? Specific examples of innovative methodologies (e.g., digital health tools, community-based interventions) would make this recommendation more

---

## [Decision Letter · Decision Letter 1]

25 Jun 2025

PONE-D-24-44704R1Estimating HIV incidence and assessing associated risk factors among adults: Evidence from the 2018-2022 HIV vaccine preparedness cohort in Masaka, UgandaPLOS ONE

Dear Dr. Kusemererwa,

Thank you for submitting your manuscript to PLOS ONE. After careful consideration, we feel that it has merit but does not fully meet PLOS ONE’s publication criteria as it currently stands. Therefore, we invite you to submit a revised version of the manuscript that addresses the points raised during the review process.

We look forward to receiving your revised manuscript.

Kind regards,

Fred Nuwaha

Academic Editor

PLOS ONE

Additional Editor Comments:

The manuscript has been reviewed. Please respond to comments raised by the reviewers.

Reviewers' comments:

Reviewer's Responses to Questions

**Comments to the Author**

1. If the authors have adequately addressed your comments raised in a previous round of review and you feel that this manuscript is now acceptable for publication, you may indicate that here to bypass the “Comments to the Author” section, enter your conflict of interest statement in the “Confidential to Editor” section, and submit your "Accept" recommendation.

Reviewer #1: (No Response)

Reviewer #2: All comments have been addressed

2. Is the manuscript technically sound, and do the data support the conclusions?

Reviewer #1: Partly

Reviewer #2: Yes

3. Has the statistical analysis been performed appropriately and rigorously?

Reviewer #1: No

Reviewer #2: Yes

4. Have the authors made all data underlying the findings in their manuscript fully available?

Reviewer #1: Yes

Reviewer #2: Yes

5. Is the manuscript presented in an intelligible fashion and written in standard English?

Reviewer #1: Yes

Reviewer #2: Yes

6. Review Comments to the Author

Reviewer #1: Reviewing of the revised version of paper titled

Estimating HIV incidence and assessing associated risk factors among adults : Evidence from the 2018-2022 HIV vaccine preparedness cohort in Masaka, Uganda.

Thank you again for the opportunity to review the revised version of this paper, which address a study with the potential to make a significant contribution to HIV research, particularly within the framework of the "First 95" initiative. I would like to thank the authors for the responses provided. Overall, most of my comments have been integrated, and the quality of the manuscript has improved accordingly. Nonetheless, some comments issues remain (particularly on methodological aspects) that need further attention from the authors to better improve quality of the work. I also strongly recommend that in their response to reviewer comments, the authors systematically reference the specific line numbers of the revised manuscript. This would greatly facilitate the assessment of their revisions and allow a smoother review process. Below are my additional comments following the authors’ responses.

Lines 154 – 160: Thank you for these clarifications on my previous comment 5 and the additional details added in the manuscript regarding the screening algorithm and quality control of the HIV testing procedures. My concern was related in the fact that the national algorithm, although highly specific and sensitive, relies on rapid diagnostic tests that target antibodies, which may be present at low quantity during the early stages of infection (i.e., during seroconversion). Consequently, there remains a small probability that participants in the acute phase of infection could have been misclassified as HIV-negative at baseline and inadvertently included in the cohort. It would therefore be appropriate to acknowledge this as a potential limitation in the discussion section.

Lines 180 – 182: Thank you to the authors for their clarifications following my previous Comment 6. However, I respectfully disagree with the rationale provided, particularly the argument stating:

“The analytical approach used in this paper is a standard method for analyzing incidence data from a single study site. Similar approaches have been applied in other studies (Ahmed, Lutalo et al. 2001, De Schacht, Mabunda et al. 2014, Westreich, Jamal et al. 2014, Gómez-Olivé, Houle et al. 2020). If this were a multi-site study or one where the same individual was assessed multiple times for an outcome (e.g., repeated blood pressure measurements), incorporating random effects or using a GEE model would be appropriate. However, in our case, each participant had a single outcome (HIV status at the end of follow-up), and the study was conducted at a single site (Masaka). Therefore, there was no intra-individual variability or clear random effect to account for.”

This reasoning is not convincing for the following reasons:

1. The fact that the study was conducted at a single site does not in itself justify ignoring intra-individual variability. Even within a single site, a cohort of participants followed over time inevitably induces within-subject correlation across certain characteristics to each individual.

2. Similarly, the fact that each participant had a single binary outcome (HIV status) does not justify ignoring intra-individual variability, as several covariates (such as age, occupation, education level, marital status, and condom use at last sex (past 3 months)) may vary over time for a single participant and have been collected repeatedly throughout follow-up.

3. The statement, “If this were a multi-site study or one where the same individual was assessed multiple times for an outcome [...] incorporating random effects or using a GEE model would be appropriate”, appears to contradict the study design and reported results. According to the manuscript (line 104), participants were followed from 18 July 2018 to 31 December 2022. During this period, the same participants were followed over time, and several key characteristics (such as age, occupation, education level, marital status, residence, abnormal genital discharge in the past 3 months, and condom use) may have changed for a given participant over the four-year follow-up. This could be observed in the distribution of participant characteristics presented in Tables 1 and 2 (lines 266 and 289, respectively). Therefore, time-varying changes in these factors at the individual level could significantly influence HIV infection risk.

In light of the above, using a Poisson model with random effects or a marginal Poisson model using Generalized Estimating Equations (GEE) (with a random intercept and interaction terms with time for key variables) is both justified and methodologically appropriate for this study. Even if the results from this model was to remain similar to those currently presented, from a methodological standpoint, this approach better accounts for the longitudinal nature of the data and should be seriously considered by the authors.

Lines 154 – 160: Thank you to the authors for the clarifications provided in response to my previous Comment 7 regarding the sensitivity analysis in light of the high loss to follow-up rate (22%). For the sensitivity analyses, the authors may formulate assumptions (ideally informed by a review of the literature) and present the corresponding results. A plausible assumption, in the lack of evidence to suggest otherwise, would be that the HIV incidence among participants lost to follow-up is similar to that observed among those who completed follow-up.

Lines 210 – 215: Thank you for these clarifications following my previous Comment 8 and for the additional information provided in the footnote of Table 1. However, I am somewhat confused by the description: “§Other includes (multiple options allowed): Professional/technical worker (19%), sales/service worker (24%), subsistence agricultural worker (16%), craft and related trades worker (15%), house helper/labourer (16%), motorcyclist (8%), unemployed (7%), etc.; N/A, not applicable.” The sum of these percentages is 105%, and this is without considering the "etc." category. Could there be an error in these calculations? If so, please correct accordingly.

Additionally, reporting the HIV incidence within each of these subgroups would be valuable for readers.

Lines 161 – 162: Thank you to the authors for including the HIV diagnostic algorithm (S1 Fig). However, after reviewing it, I found the flow diagram somewhat confusing, particularly regarding the classification of positive and inconclusive cases. According to the algorithm, a participant is declared HIV-positive after reactive results on both the Determine and Stat-Pak tests. However, there is an arrow that subsequently leads to a "reactive" outcome followed by an "inconclusive" classification. This appears to be a misalignment or error in the linkage of results and arrows within the figure. I kindly ask the authors to review and correct this accordingly.

In addition, I would like to raise the following questions for clarification:

• Is the tie-breaker test (SD Bioline) routinely supplied by the Ministry of Health and available in all health facilities to ensure consistent implementation of this algorithm?

• As part of quality assurance procedures, is a second laboratory technician required to independently repeat the algorithm for a participant before an HIV-positive result is confirmed?

It would be helpful if the authors could clarify and include these details in the manuscript.

Lines 210-218: Thank you to the authors for including Supporting Table 2 in response to my previous Comment 27. As previously noted in table 1 (Line 266), the total percentage exceeds 100% for the “Other” occupation category: “¶Other includes (multiple options allowed): Professional/technical worker (19%), sales/service worker (24%), subsistence agricultural worker (16%), craft and related trades worker (15%), house helper/labourer (16%), motorcyclist (8%), unemployed (7%), etc.”

Please review and correct this accordingly.

Line 244 Fig 1: Thank you for this clarification regarding my previous Comment 29, which reads as follows:

“We have provided a detail to distinguish between the cohort and the trial in the methods section as written below: The main aim of the cohort study was to identify HIV-negative adults at high risk of HIV infection and prepare them for participation in a phase IIb HIV prophylactic vaccine trial (NCT04066881) at sites in Mozambique, South Africa, Tanzania, and Uganda. The trial aimed to determine the efficacy of two HIV prophylactic vaccine regimens and to compare the effectiveness of Truvada and Descovy as PrEP (Joseph, Kaleebu et al. 2019).”

Does this imply that the 30 HIV-negative participants who completed the cohort study but were not be enrolled in the PrEPVacc trial were no longer considered to be at high risk of HIV infection? If so, this seems somewhat surprising, given that being at high risk of HIV acquisition was an inclusion criterion for enrollment in the cohort study.

I recommend that the authors add a footnote to Figure 1 to clarify this subgroup (N = 30) and explain the reason for their non-enrollment in the PrEPVacc trial.

Reviewer #2: The authors have thorougly adressed all the comments. The current version meets the publication expectation and I recommend approval

7. PLOS authors have the option to publish the peer review history of their article (what does this mean?). If published, this will include your full peer review and any attached files.

Reviewer #1: **Yes:** Cyprien KENGNE NDE

Reviewer #2: **Yes:** Dathan M Byonanebye

---

## [Author Response · Author response to Decision Letter 2]

30 Jul 2025

Response sheet for submission ID PONE-D-24-44704 R2: Estimating HIV incidence and assessing associated risk factors among adults: Evidence from the 2018-2022 HIV vaccine preparedness cohort in Masaka, Uganda

Comments and responses

We wish to immensely thank the reviewers for their valuable comments. Please find below a point-by-point response to the new set of comments raised.

Reviewer #1

Thank you again for the opportunity to review the revised version of this paper, which addresses a study with the potential to make a significant contribution to HIV research, particularly within the framework of the "First 95" initiative. I would like to thank the authors for the responses provided. Overall, most of my comments have been integrated, and the quality of the manuscript has improved accordingly. Nonetheless, some comments issues remain (particularly on methodological aspects) that need further attention from the authors to better improve the quality of the work. I also strongly recommend that in their response to reviewer comments, the authors systematically reference the specific line numbers of the revised manuscript. This would greatly facilitate the assessment of their revisions and allow a smoother review process. Below are my additional comments following the authors’ responses.

Comment 1: Lines 154 – 160: Thank you for these clarifications on my previous comment 5 and the additional details added in the manuscript regarding the screening algorithm and quality control of the HIV testing procedures. My concern was related in the fact that the national algorithm, although highly specific and sensitive, relies on rapid diagnostic tests that target antibodies, which may be present at low quantity during the early stages of infection (i.e., during seroconversion). Consequently, there remains a small probability that participants in the acute phase of infection could have been misclassified as HIV-negative at baseline and inadvertently included in the cohort. It would therefore be appropriate to acknowledge this as a potential limitation in the discussion section.

Response: We thank the reviewer for their insight. We have included this suggestion as a limitation in lines 358-361.

Comment 2: Lines 180 – 182: Thank you to the authors for their clarifications following my previous Comment 6. However, I respectfully disagree with the rationale provided, particularly the argument stating:

“The analytical approach used in this paper is a standard method for analyzing incidence data from a single study site. Similar approaches have been applied in other studies (Ahmed, Lutalo et al. 2001, De Schacht, Mabunda et al. 2014, Westreich, Jamal et al. 2014, Gómez-Olivé, Houle et al. 2020). If this were a multi-site study or one where the same individual was assessed multiple times for an outcome (e.g., repeated blood pressure measurements), incorporating random effects or using a GEE model would be appropriate. However, in our case, each participant had a single outcome (HIV status at the end of follow-up), and the study was conducted at a single site (Masaka). Therefore, there was no intra-individual variability or clear random effect to account for.”

This reasoning is not convincing for the following reasons:

1. The fact that the study was conducted at a single site does not in itself justify ignoring intra-individual variability. Even within a single site, a cohort of participants followed over time inevitably induces within-subject correlation across certain characteristics to each individual.

2. Similarly, the fact that each participant had a single binary outcome (HIV status) does not justify ignoring intra-individual variability, as several covariates (such as age, occupation, education level, marital status, and condom use at last sex (past 3 months)) may vary over time for a single participant and have been collected repeatedly throughout follow-up.

3. The statement, “If this were a multi-site study or one where the same individual was assessed multiple times for an outcome [...] incorporating random effects or using a GEE model would be appropriate”, appears to contradict the study design and reported results. According to the manuscript (line 104), participants were followed from 18 July 2018 to 31 December 2022. During this period, the same participants were followed over time, and several key characteristics (such as age, occupation, education level, marital status, residence, abnormal genital discharge in the past 3 months, and condom use) may have changed for a given participant over the four-year follow-up. This could be observed in the distribution of participant characteristics presented in Tables 1 and 2 (lines 266 and 289, respectively). Therefore, time-varying changes in these factors at the individual level could significantly influence HIV infection risk.

In light of the above, using a Poisson model with random effects or a marginal Poisson model using Generalized Estimating Equations (GEE) (with a random intercept and interaction terms with time for key variables) is both justified and methodologically appropriate for this study. Even if the results from this model was to remain similar to those currently presented, from a methodological standpoint, this approach better accounts for the longitudinal nature of the data and should be seriously considered by the authors.

Response: We thank the reviewer for these comments. We agree with your thoughts on this. The change in covariates (e.g demographics, risk behavior) over time can influence an individual’s risk of HIV infection over time. However, we are unsure as to whether the model you recommend is suitable for this analysis. It seems as though the suggested analysis would be suitable if there was intra-individual variability in the outcome of interest (HIV status in our case), not just intra-individual variability in the predictors (age, other demographics, risk behaviors etc.). Any publications investigating HIV incidence that used a similar approach to the one suggested may assist with this.

Alternatively, an analytical approach that would address your concerns would be incorporating time varying predictors in the survival analysis model used in our study. (Of note, we conducted survival analysis, assuming an exponential distribution - this is synonymous with “Poisson regression”). However, our study had a limited number of person years and events [For reference: Table 1 in the manuscript includes person years and HIV incident cases by the year of follow-up (First year, Second year and third year)]. Considering the limited data that was available beyond 1 year of follow-up, more detailed analyses investigating the impact of change in participant characteristics and behaviors over time on HIV incidence, are substantially underpowered/ limited. Additionally, participants had varying times of follow-up with some as little as 3 months, and others as long as 2 years or more, with only risk behaviour data collected every 6 months (other data were only collected at baseline). In consideration of the limited data, we opted to only conduct two main multi-variable analyses, one investigating the impact of baseline risk behaviors (adjusted for other predictors) and the other investigating the impact of the risk behavior data reported closest to the time-of sero-conversion or end of follow-up (also adjusted for other characteristics).

Comment 3: Lines 154 – 160: Thank you to the authors for the clarifications provided in response to my previous Comment 7 regarding the sensitivity analysis in light of the high loss to follow-up rate (22%). For the sensitivity analyses, the authors may formulate assumptions (ideally informed by a review of the literature) and present the corresponding results. A plausible assumption, in the lack of evidence to suggest otherwise, would be that the HIV incidence among participants lost to follow-up is similar to that observed among those who completed follow-up.

Response: We thank the reviewer for this comment. We acknowledged in the revised manuscript in lines 344-348 that “A major limitation of this study is that 22% of individuals did not attend any follow-up visits. These individuals were more likely to be young (<25years), female, recreational drug users, and persons who engage in transactional sex, subpopulations that are also associated with a higher risk of HIV acquisition. This likely introduced selection bias, potentially leading to an underestimation of HIV incidence in our study”.

Comment 4: Lines 210 – 215: Thank you for these clarifications following my previous Comment 8 and for the additional information provided in the footnote of Table 1. However, I am somewhat confused by the description: “§Other includes (multiple options allowed): Professional/technical worker (19%), sales/service worker (24%), subsistence agricultural worker (16%), craft and related trades worker (15%), house helper/labourer (16%), motorcyclist (8%), unemployed (7%), etc.; N/A, not applicable.” The sum of these percentages is 105%, and this is without considering the "etc." category. Could there be an error in these calculations? If so, please correct accordingly.

Additionally, reporting the HIV incidence within each of these subgroups would be valuable for readers.

Response: We thank the reviewer for this comment. The percentages add up to more than 100% because participants could list more than one occupation and be counted in more than one category. We have now added a supplementary table (S3 Tab) with HIV incidence in each of the different occupations.

Comment 5: Lines 161 – 162: Thank you to the authors for including the HIV diagnostic algorithm (S1 Fig). However, after reviewing it, I found the flow diagram somewhat confusing, particularly regarding the classification of positive and inconclusive cases. According to the algorithm, a participant is declared HIV-positive after reactive results on both the Determine and Stat-Pak tests. However, there is an arrow that subsequently leads to a "reactive" outcome followed by an "inconclusive" classification. This appears to be a misalignment or error in the linkage of results and arrows within the figure. I kindly ask the authors to review and correct this accordingly.

In addition, I would like to raise the following questions for clarification:

• Is the tie-breaker test (SD Bioline) routinely supplied by the Ministry of Health and available in all health facilities to ensure consistent implementation of this algorithm?

• As part of quality assurance procedures, is a second laboratory technician required to independently repeat the algorithm for a participant before an HIV-positive result is confirmed?

It would be helpful if the authors could clarify and include these details in the manuscript.

Response: We appreciate the reviewer’s observation regarding the figure misalignment and have corrected it accordingly.

We confirm that SD-Bioline serves as the designated tie-breaker test, as routinely provided by the Ministry of Health. It is readily available in health facilities across the country.

While the testing algorithm was not repeated by a second laboratory technician, this step is not required under the current national HIV testing algorithm. Nevertheless, to ensure quality, HIV testing procedures were performed by certified HIV testing service counsellors. Internal quality control (IQC) procedures were performed weekly using known positive and negative samples, while external quality assurance (EQA) assessments were conducted quarterly under the Uganda Virus Research Institute’s national reference laboratory EQA proficiency testing scheme (lines 164-168).

Comment 6: Lines 210-218: Thank you to the authors for including Supporting Table 2 in response to my previous Comment 27. As previously noted in table 1 (Line 266), the total percentage exceeds 100% for the “Other” occupation category: “¶Other includes (multiple options allowed): Professional/technical worker (19%), sales/service worker (24%), subsistence agricultural worker (16%), craft and related trades worker (15%), house helper/labourer (16%), motorcyclist (8%), unemployed (7%), etc.”

Please review and correct this accordingly.

Response: We thank the reviewer for their keen eye. The footnote in the supporting table 2 has been edited to remove the percentages as they didn’t belong to this table.

Comment 7: Line 244 Fig 1: Thank you for this clarification regarding my previous Comment 29, which reads as follows:

“We have provided a detail to distinguish between the cohort and the trial in the methods section as written below: The main aim of the cohort study was to identify HIV-negative adults at high risk of HIV infection and prepare them for participation in a phase IIb HIV prophylactic vaccine trial (NCT04066881) at sites in Mozambique, South Africa, Tanzania, and Uganda. The trial aimed to determine the efficacy of two HIV prophylactic vaccine regimens and to compare the effectiveness of Truvada and Descovy as PrEP (Joseph, Kaleebu et al. 2019).”

Does this imply that the 30 HIV-negative participants who completed the cohort study but were not enrolled in the PrEPVacc trial were no longer considered to be at high risk of HIV infection? If so, this seems somewhat surprising, given that being at high risk of HIV acquisition was an inclusion criterion for enrollment in the cohort study.

I recommend that the authors add a footnote to Figure 1 to clarify this subgroup (N = 30) and explain the reason for their non-enrollment in the PrEPVacc trial.

Response: The participants’ “risk of HIV infection” was re-evaluated at the time of screening for the trial, and they were considered to no longer be at elevated risk of infection as per their reported risk behaviours, or PrEP uptake status. A footnote has been added in Lines 240-242.

Notably, participation in a study that included regular HIV testing, quarterly counselling, and access to prevention information may have influenced participants’ sexual risk behaviours. Similar studies, including that involving this cohort, have reported declines in risky behaviour over time, likely due to consistent counselling (1, 2).

References

1. Kitonsa J, Kansiime S, Kusemererwa S, Onyango M, Nayiga B, Kabarambi A, Mugisha JO, Kaleebu P, Ruzagira E. Changes in self-reported risky sexual behaviour indicators among adults receiving regular risk reduction counselling and optional initiation of pre-exposure prophylaxis in an HIV vaccine preparedness study in Masaka, Uganda. Global Health Action. 2023 Dec 31;16(1):2242672.

2. Faini D, Hanson C, Baisley K, Kapiga S, Hayes R. Sexual behaviour, changes in sexual behaviour and associated factors among women at high risk of HIV participating in feasibility studies for prevention trials in Tanzania. PloS one. 2020 Apr 16;15(4):e0231766.

---

## [Decision Letter · Decision Letter 2]

3 Feb 2026

PONE-D-24-44704R2Estimating HIV incidence and assessing associated risk factors among adults: Evidence from the 2018-2022 HIV vaccine preparedness cohort in Masaka, UgandaPLOS One

Dear Dr. Kusemererwa,

Thank you for submitting your manuscript to PLOS ONE. After careful consideration, we feel that it has merit but does not fully meet PLOS ONE’s publication criteria as it currently stands. Therefore, we invite you to submit a revised version of the manuscript that addresses the points raised during the review process.

Please submit your revised manuscript by Mar 20 2026 11:59PM. If you will need more time than this to complete your revisions, please reply to this message or contact the journal office at plosone@plos.org. Please include the following items when submitting your revised manuscript:

We look forward to receiving your revised manuscript.

Kind regards,

Hamufare Dumisani Mugauri, Ph.D. Epidemiology and Public Health

Academic Editor

PLOS One

Journal Requirements:

Reviewers' comments:

Reviewer's Responses to Questions

**Comments to the Author**

1. If the authors have adequately addressed your comments raised in a previous round of review and you feel that this manuscript is now acceptable for publication, you may indicate that here to bypass the “Comments to the Author” section, enter your conflict of interest statement in the “Confidential to Editor” section, and submit your "Accept" recommendation.

Reviewer #1: (No Response)

Reviewer #2: All comments have been addressed

2. Is the manuscript technically sound, and do the data support the conclusions?

Reviewer #1: Partly

Reviewer #2: Yes

3. Has the statistical analysis been performed appropriately and rigorously?

Reviewer #1: No

Reviewer #2: Yes

4. Have the authors made all data underlying the findings in their manuscript fully available?

Reviewer #1: No

Reviewer #2: Yes

5. Is the manuscript presented in an intelligible fashion and written in standard English?

Reviewer #1: Yes

Reviewer #2: Yes

6. Review Comments to the Author

Reviewer #1: Reviewing of the revised version of paper titled: Estimating HIV incidence and assessing associated risk factors among adults : Evidence from the 2018-2022 HIV vaccine preparedness cohort in Masaka, Uganda.

Thank you once again for the opportunity to review the second revised version of this manuscript, which addresses an important topic with strong potential to advance HIV research, particularly in the context of the “First 95” initiative. I appreciate the authors’ efforts in revising the paper and for the detailed responses provided. I acknowledge the authors’ efforts and the improvements made to the manuscript in response to earlier comments.

That said, despite progress, some key issues remain insufficiently addressed, most notably regarding methodological clarity and statistical rigor, as well as the presentation and consistency of the results in certain tables. These issues are critical and must be resolved to ensure the accuracy, and scientific validity of the findings.

Below, I provide further comments and recommendations that require authors’ attention.

Lines 266–270: Thank you for these additional details regarding my previous Comment 2, which concerns a fundamental methodological issue I have raised from the beginning of this manuscript revision process in my earlier review reports. I am therefore returning to it for the third time!

Regarding the authors’ response: “We agree with your thoughts on this. The change in covariates (e.g., demographics, risk behavior) over time can influence an individual’s risk of HIV infection over time. However, we are unsure as to whether the model you recommend is suitable for this analysis. It seems as though the suggested analysis would be suitable if there was intra-individual variability in the outcome of interest (HIV status in our case), not just intra-individual variability in the predictors (age, other demographics, risk behaviors etc.). Any publications investigating HIV incidence that used a similar approach to the one suggested may assist with this”:

• The authors should consider that in a random-effects Poisson model, it models the probability for a participant to have a given value of the outcome (here, HIV status: positive or negative). Thus, a participant might remain negative throughout the cohort follow-up, but their probability of being positive varies due to changes in other covariates for that same participant and other participants.

• There is ample evidence in the literature supporting this approach; a few references examples[1,2] are provided below

1 Laurent C, Yaya I, Cuer B, et al. Human Immunodeficiency Virus Seroconversion Among Men Who Have Sex With Men Who Use Event-Driven or Daily Oral Pre-Exposure Prophylaxis (CohMSM-PrEP): A Multi-Country Demonstration Study From West Africa. Clin Infect Dis Off Publ Infect Dis Soc Am. 2023;77:606–14. doi: 10.1093/cid/ciad221

2 M M, A B, Rm G, et al. Longitudinal population-level HIV epidemiologic and genomic surveillance highlights growing gender disparity of HIV transmission in Uganda. Nat Microbiol. 2024;9. doi: 10.1038/s41564-023-01530-8

Concerning the authors’ further arguments: “Alternatively, an analytical approach that would address your concerns would be incorporating time-varying predictors in the survival analysis model used in our study …… (also adjusted for other characteristics)” :

this does not justify the a priori exclusion of using a Poisson model with random effects or a marginal Poisson model using Generalized Estimating Equations (GEE) which is the appropriate statistical approach for this study design, because the claimed lack of sufficient data / limited power beyond the first year of follow-up should be demonstrated by presenting estimates obtained from this model applied to the collected data, rather than simply being assumed.

Additionally, the statement: “Of note, we conducted survival analysis, assuming an exponential distribution – this is synonymous with Poisson regression” is misleading. While it is true that an exponential survival model (assuming a constant hazard) can produce estimates equivalent to a Poisson regression of events per person-time under certain conditions, the two approaches are not strictly synonymous. Poisson regression is fundamentally a model for count data (ideal for this study analysis), whereas the exponential survival model is a parametric time-to-event model. Equivalence only holds under the assumption of a constant hazard and when person-time is correctly incorporated as an offset in the Poisson model. Furthermore, this was not mentioned in the methodology section of the manuscript, which raises concern. So the authors should clearly clarify the statistical modelling approach actually used, and ensure it is accurately described in the methodology section of the paper.

Lines 358–361: Thank you to the authors for addressing my previous comment #1 in the revised R2 version of the manuscript. However, the clarification remains insufficient. The manuscript still does not clearly articulate the limitation related to the potential bias of inclusion participants already infected (in seroconversion) but misclassified as HIV-negative due to the reliance on the national antibody-based rapid testing algorithm. The authors should explicitly acknowledge this limitation of selection bias of participants and discuss its implications, given the fact that additional HIV tests were performed on participants quarterly after enrollment. This point could also be described in a separate paragraph, as it differs from the previously presented limitations in this section.

Lines 266–270: Thank you to the authors for these clarifications regarding my previous Comment #4 and for adding Supplementary Table 3 (S3 Table). However, the approach of allowing a participant to fall into multiple sub-categories within the “Other” occupation category and presenting the results as such raises concerns regarding interpretation for at least three reasons:

1. The total count for the “Other” category would then be at least 354 (i.e., 105% when summing the stated percentages without accounting for “etc.”), instead of 337. Consequently, the overall sample size would become at least 1,132 instead of 1,115. This artificially inflates the number of participants effectively enrolled and having completed at least one visit.

2. A single participant could contribute simultaneously to multiple sub-categories of the “Other” occupation category. This introduces ambiguity in interpreting incidence estimates. For example, if the participant identified as new HIV case (seroconverting) under the “House helper/labourer” category (S3 Table) is the same individual identified as new HIV case (seroconverting) under the “Sales/service worker” category because he selected the both category, this would inflate the incidence estimate for the “Other” occupation category.

3. This approach violates an underlying assumption of the statistical model used in the analysis, which stipulate that observations (participants) are independent and identically distributed (i.i.d).

In light of the above, I suggest that the authors should:

1. Classify each participant into a single occupation category (for example, by defining a rule such as selecting the first option chosen or the primary occupation recorded in the database) and specify the methodology used in the appropriate section of the paper;

2. Specify the number and percentage for each sub-category in S3 Tab within “Other” (e.g., Professional/technical worker: 66 (19%)) and ensure that the total count equals 337 and that percentages sum to 100%;

3. Adjust the participant counts and percentages for these sub-categories in the footnotes of Table 1 (line 268);

4. Refer the reader to S3 Table for additional details. In the current manuscript, no reference to S3 Table is made in the interpretation of Table 1 (lines 250–263).

S2 Tab: Thank you to the authors for addressing my previous Comment 6. However, the footnote reference “§ Formal/informal employment, family, spouse etc.” is ambiguous, as the symbol “§” does not appear anywhere in the S2 Tab. The authors should insert the symbol in the appropriate location within the table before referencing it in the footnote, or remove the footnote if it is not necessary.

Lignes 266-270 and 289-292:

1. The footnote in Tables 1 and 2 (“n, number of HIV infections;”) is ambiguous, as both tables contain two columns labeled “n (%)” and “n.” However, these two columns do not refer to the same information. I recommend that the authors correct this to avoid confusion.

2. In addition, the authors should provide clarification on the points below:

a. The “n (%)” total is identical in both tables (1115 (100%)), although Table 1 presents baseline characteristics and Table 2 presents characteristics at the end of cohort follow-up. Given that the authors reported a 22% loss to follow-up in the cohort, one would expect the sample size to decrease in Table 2. What explains this?

b. The variable “Gender,” which was statistically significant in Table 1, does not appear in Table 2. However, the lines 194-197 clearly state that “Age and gender were included in the multivariable models a priori based on established evidence from previous research, which consistently identifies them as major key risk factors for incident HIV [26-28].” Why, then, is gender absent from Table 2?

c. Each variable in the “n (%)” column of Tables 1 and 2 sums to 1115 (100%), which suggests that there were no missing values (a scenario highly unlikely in cohort studies) or that missing values were imputed. If imputation was performed, the authors should specify the imputation methods used in the methodology section (statistical analysis). Could the authors clarify this!

Reviewer #2: All comments have been adressed. I thank the authors for addressing all the comments. The revised paper reads much better and I recommend approval.

7. PLOS authors have the option to publish the peer review history of their article (what does this mean?). If published, this will include your full peer review and any attached files.

Reviewer #1: **Yes:** Cyprien Kengne-Nde

Reviewer #2: **Yes:** Byonanebye M. Dathan

---

## [Author Response · Author response to Decision Letter 3]

20 Mar 2026

Response sheet for Revision 3 of the manuscript entitled Estimating HIV incidence and assessing associated risk factors among adults: Evidence from the 2018-2022 HIV vaccine preparedness cohort in Masaka, Uganda.

Comment 1: Thank you once again for the opportunity to review the second revised version of this manuscript, which addresses an important topic with strong potential to advance HIV research, particularly in the context of the “First 95” initiative. I appreciate the authors’ efforts in revising the paper and for the detailed responses provided. I acknowledge the authors’ efforts and the improvements made to the manuscript in response to earlier comments.

That said, despite progress, some key issues remain insufficiently addressed, most notably regarding methodological clarity and statistical rigor, as well as the presentation and consistency of the results in certain tables. These issues are critical and must be resolved to ensure the accuracy and scientific validity of the findings.

Below, I provide further comments and recommendations that require authors’ attention.

Response: We thank the reviewer for their interest in our manuscript and constructive comments. We appreciate the additional insights provided which have helped us improve the overall quality of the manuscript.

Comment 2: Lines 266–270: Thank you for these additional details regarding my previous Comment 2, which concerns a fundamental methodological issue I have raised from the beginning of this manuscript revision process in my earlier review reports. I am therefore returning to it for the third time!

Regarding the authors’ response: “We agree with your thoughts on this. The change in covariates (e.g., demographics, risk behavior) over time can influence an individual’s risk of HIV infection over time. However, we are unsure as to whether the model you recommend is suitable for this analysis. It seems as though the suggested analysis would be suitable if there was intra-individual variability in the outcome of interest (HIV status in our case), not just intra-individual variability in the predictors (age, other demographics, risk behaviors etc.). Any publications investigating HIV incidence that used a similar approach to the one suggested may assist with this”:

• The authors should consider that in a random-effects Poisson model, it models the probability for a participant to have a given value of the outcome (here, HIV status: positive or negative). Thus, a participant might remain negative throughout the cohort follow-up, but their probability of being positive varies due to changes in other covariates for that same participant and other participants.

• There is ample evidence in the literature supporting this approach; a few references examples[1,2] are provided below

1 Laurent C, Yaya I, Cuer B, et al. Human Immunodeficiency Virus Seroconversion Among Men Who Have Sex With Men Who Use Event-Driven or Daily Oral Pre-Exposure Prophylaxis (CohMSM-PrEP): A Multi-Country Demonstration Study From West Africa. Clin Infect Dis Off Publ Infect Dis Soc Am. 2023;77:606–14. doi: 10.1093/cid/ciad221

2 M M, A B, Rm G, et al. Longitudinal population-level HIV epidemiologic and genomic surveillance highlights growing gender disparity of HIV transmission in Uganda. Nat Microbiol. 2024;9. doi: 10.1038/s41564-023-01530-8

Concerning the authors’ further arguments: “Alternatively, an analytical approach that would address your concerns would be incorporating time-varying predictors in the survival analysis model used in our study …… (also adjusted for other characteristics)” : this does not justify the a priori exclusion of using a Poisson model with random effects or a marginal Poisson model using Generalized Estimating Equations (GEE) which is the appropriate statistical approach for this study design, because the claimed lack of sufficient data / limited power beyond the first year of follow-up should be demonstrated by presenting estimates obtained from this model applied to the collected data, rather than simply being assumed.

Additionally, the statement: “Of note, we conducted survival analysis, assuming an exponential distribution – this is synonymous with Poisson regression” is misleading. While it is true that an exponential survival model (assuming a constant hazard) can produce estimates equivalent to a Poisson regression of events per person-time under certain conditions, the two approaches are not strictly synonymous. Poisson regression is fundamentally a model for count data (ideal for this study analysis), whereas the exponential survival model is a parametric time-to-event model. Equivalence only holds under the assumption of a constant hazard and when person-time is correctly incorporated as an offset in the Poisson model. Furthermore, this was not mentioned in the methodology section of the manuscript, which raises concern. So the authors should clearly clarify the statistical modelling approach actually used, and ensure it is accurately described in the methodology section of the paper.

Response: Thank you for these very insightful comments and the publications. We have now included a supplementary analysis (Supporting table 4) of the “Risk indicators association with HIV acquisition … (accounting for repeated assessments of risk at individual level over time)”. We utilised Mixed-effects Poisson Regression models with individual-specific random effects for that analysis.

Also, in the methodology section of the paper, we have now acknowledged that, “The natural log of the person-time (years) of follow-up was incorporated as an offset in the Poisson regression models”.

Comment 3: Lines 358–361: Thank you to the authors for addressing my previous comment #1 in the revised R2 version of the manuscript. However, the clarification remains insufficient. The manuscript still does not clearly articulate the limitation related to the potential bias of inclusion participants already infected (in seroconversion) but misclassified as HIV-negative due to the reliance on the national antibody-based rapid testing algorithm. The authors should explicitly acknowledge this limitation of selection bias of participants and discuss its implications, given the fact that additional HIV tests were performed on participants quarterly after enrollment. This point could also be described in a separate paragraph, as it differs from the previously presented limitations in this section.

Response: We appreciate the reviewer for their suggestion. This limitation of using antibody-based rapid diagnostic tests has been clarified and rewritten as a separate paragraph which now reads as follows: HIV infection was diagnosed using antibody-based rapid diagnostic tests, which may not identify very recent infections, before seroconversion has occurred, potentially resulting in delayed diagnosis and misclassification of HIV status. Quarterly HIV testing substantially mitigated this risk by increasing the likelihood of detecting new HIV infections during follow-up, although some underestimation of HIV incidence may still have occurred. Lines 379-383

Comment 4: Lines 266–270: Thank you to the authors for these clarifications regarding my previous Comment #4 and for adding Supporting Table 3 (S3 Table). However, the approach of allowing a participant to fall into multiple sub-categories within the “Other” occupation category and presenting the results as such raises concerns regarding interpretation for at least three reasons:

1. The total count for the “Other” category would then be at least 354 (i.e., 105% when summing the stated percentages without accounting for “etc.”), instead of 337. Consequently, the overall sample size would become at least 1,132 instead of 1,115. This artificially inflates the number of participants effectively enrolled and having completed at least one visit.

2. A single participant could contribute simultaneously to multiple sub-categories of the “Other” occupation category. This introduces ambiguity in interpreting incidence estimates. For example, if the participant identified as new HIV case (seroconverting) under the “House helper/labourer” category (S3 Table) is the same individual identified as new HIV case (seroconverting) under the “Sales/service worker” category because he selected the both category, this would inflate the incidence estimate for the “Other” occupation category.

3. This approach violates an underlying assumption of the statistical model used in the analysis, which stipulate that observations (participants) are independent and identically distributed (i.i.d).

In light of the above, I suggest that the authors should:

1. Classify each participant into a single occupation category (for example, by defining a rule such as selecting the first option chosen or the primary occupation recorded in the database) and specify the methodology used in the appropriate section of the paper;

2. Specify the number and percentage for each sub-category in S3 Tab within “Other” (e.g., Professional/technical worker: 66 (19%)) and ensure that the total count equals 337 and that percentages sum to 100%;

3. Adjust the participant counts and percentages for these sub-categories in the footnotes of Table 1 (line 268);

4. Refer the reader to S3 Table for additional details. In the current manuscript, no reference to S3 Table is made in the interpretation of Table 1 (lines 250–263).

Response: Thank you for these suggestions.

For the main analyses, including the regression models, participants were categorized into only four distinct occupation groups: (i) Sex workers (n=509), ii) Salon/bar/lodge workers (n=181), iii) Subsistence fisheries workers (88) and iv) Other occupations, (n=337) as shown in Table 1. The overall sample size was therefore 1,115 participants.

Because the number of participants in the ‘Other’ group category is large, Supplementary Table 2 and the footnotes of Table 1 provide further details of the occupations in this category. These comprise Professional/technical worker (64, 19%), sales/service worker (80, 24%), subsistence agricultural worker (55, 16%), craft and related trades worker (49, 15%), house helper/labourer (53, 16%), motorcyclist (26, 8%), unemployed individuas (24, 7%), and other occupations not listed (18, 15%). Participants were able to report more than one occupation; therefore, the total responses exceed 337, which is the total number of study participants in this category. Within the ‘Other’ group, each of the three HIV incident cases is unique and counted only in one occupational category; therefore, there is no risk of inflating the incidence estimate for any of these categories.

For the main analysis (Table 1), the other ‘Other’ occupational category was not further disaggregated because this would result in several small groups, making the analysis and interpretation of the results unnecessarily complex.

As noted above, the disaggregated groups are presented only in Supplementary Table 2 to provide additional detail on the ‘Other’ category. An explanation is also provided for why the number of responses exceeds the number of participants. Each of the three HIV infections is unique to its respective disaggregated occupational group. Moreover, these disaggregated groups were not included in the regression models; therefore, the assumption of independence is unlikely to be violated.

We have added a reference to Supplementary Table 2 in the footer of Table 1.

Comment 5: S2 Tab: Thank you to the authors for addressing my previous Comment 6. However, the footnote reference “§ Formal/informal employment, family, spouse etc.” is ambiguous, as the symbol “§” does not appear anywhere in the S2 Tab. The authors should insert the symbol in the appropriate location within the table before referencing it in the footnote, or remove the footnote if it is not necessary.

Response: - Thank you for this observation. The symbol has been removed from the updated Table (now Supplementary Table 1).

Comment 6: Lines 266-270 and 289-292:

1. The footnote in Tables 1 and 2 (“n, number of HIV infections;”) is ambiguous, as both tables contain two columns labeled “n (%)” and “n.” However, these two columns do not refer to the same information. I recommend that the authors correct this to avoid confusion.

Response: Thank you for this comment. We have revised Tables 1 and 2 to indicate that the second column shows the number and percentage of participants in each category, while the third column shows the number of HIV infections.

2. In addition, the authors should provide clarification on the points below:

a. The “n (%)” total is identical in both tables (1115 (100%)), although Table 1 presents baseline characteristics and Table 2 presents characteristics at the end of cohort follow-up. Given that the authors reported a 22% loss to follow-up in the cohort, one would expect the sample size to decrease in Table 2. What explains this?

Response: Thank you for this comment. The n (%)” total is identical in both tables (1115) because this is the number of participants who had follow-up data i.e., attended at least one follow up visit. Table 1 presents the associations between HIV incidence and baseline risk indicators, while Table 2 presents the associations between HIV incidence and the most recent HIV risk data collected from each participant during follow-up. The 22% (n=307) refers to the participants who did not attend any follow-up visits post-enrolment and is based on the total of 1422 individuals enrolled, as reported elsewhere in the manuscript (Baseline characteristics; Figure 1). Detailed baseline characteristics for all enrollees are provided in Supplementary Table 1

b. The variable “Gender,” which was statistically significant in Table 1, does not appear in Table 2. However, the lines 194-197 clearly state that “Age and gender were included in the multivariable models a priori based on established evidence from previous research, which consistently identifies them as major key risk factors for incident HIV [26-28].” Why, then, is gender absent from Table 2?

Response: Thank you for this comment. Table 2 was intended to present the associations between HIV incidence and risk indicator data collected at the final follow-up assessment. Age, gender, residence, and occupation were included a priori as adjustment variables in the multivariable models. For presentation purposes, these variables were initially not included in Table 2 because their associations with HIV incidence were already shown in Table 1. However, we agree that presenting the adjusted estimates in these models facilitates comparsion between baseline and follow-up analyses. We have therefore now included the results for these variables in Table 2 and in Supporting 4_Table (analysis accounting for repeated assessments).

c. Each variable in the “n (%)” column of Tables 1 and 2 sums to 1115 (100%), which suggests that there were no missing values (a scenario highly unlikely in cohort studies) or that missing values were imputed. If imputation was performed, the authors should specify the imputation methods used in the methodology section (statistical analysis). Could the authors clarify this!

Response: Thank you for your comment. No imputation was performed for this analysis. The study database underwent routine data cleaning and automated data quality checks throughout data collection. Any missing or inconsistent entries generated queries that were sent to the research teams for resolution using source documents as close to real time as possible. As a result of this process, all variables included in the analyses presented in Tables 1 and 2 had complete data. If missing data had occurred, it would have been acknowledged in the footnotes of each table.

We have now clarified our data management process in the methods section. See lines 158-164.

---

## [Decision Letter · Decision Letter 3]

22 Apr 2026

Estimating HIV incidence and assessing associated risk factors among adults: Evidence from the 2018-2022 HIV vaccine preparedness cohort in Masaka, Uganda

PONE-D-24-44704R3

Dear Dr. Kusemererwa,

We’re pleased to inform you that your manuscript has been judged scientifically suitable for publication and will be formally accepted for publication once it meets all outstanding technical requirements.

Kind regards,

Hamufare Dumisani Mugauri, Ph.D. Epidemiology and Public Health

Academic Editor

PLOS One

Additional Editor Comments (optional):

Reviewers' comments:

Reviewer's Responses to Questions

**Comments to the Author**

1. If the authors have adequately addressed your comments raised in a previous round of review and you feel that this manuscript is now acceptable for publication, you may indicate that here to bypass the “Comments to the Author” section, enter your conflict of interest statement in the “Confidential to Editor” section, and submit your "Accept" recommendation.

Reviewer #3: (No Response)

Reviewer #4: All comments have been addressed

2. Is the manuscript technically sound, and do the data support the conclusions?

Reviewer #3: Yes

Reviewer #4: Yes

3. Has the statistical analysis been performed appropriately and rigorously?

Reviewer #3: I Don't Know

Reviewer #4: Yes

4. Have the authors made all data underlying the findings in their manuscript fully available?

Reviewer #3: Yes

Reviewer #4: Yes

5. Is the manuscript presented in an intelligible fashion and written in standard English?

Reviewer #3: Yes

Reviewer #4: Yes

6. Review Comments to the Author

Reviewer #3: This manuscript uses a pre-enrolment cohort to assess the risk of infection in potential participants in a PREP and vaccine trial in the Masaka district, and to describe the relationship to baseline characteristics and incidence over time of HIV risk. There is very little discussion of the incidence data, Table 2 in the results section.

The results show convincingly that risk remains tragically high in a defined group in this region, or at least did at the time of the study. A common finding in cohort studies was very strongly dictated: highest risk was found in the year after enrolment, and it declined thereafter. This might have prompted some discussion.

It appears that the paper has been very thoroughly reviewed and the authors have responded to comments appropriately.

Reviewer #4: I have looked at the comments of the previous reviewer and i am happy that all the issues that were raised have been addressed by the authors. I have only one concern. This study was conducted between 2018 and 2022. This was almost 5 years ago. Given that there are several HIV prevention technologies that have been used since then, including PrEP, do you think the HIV incidence that was estimated using these data is still valid currently?

7. PLOS authors have the option to publish the peer review history of their article (what does this mean?). If published, this will include your full peer review and any attached files.

Reviewer #3: No

Reviewer #4: No

---

## [Editor Report · Acceptance letter]

PONE-D-24-44704R3

PLOS One

Dear Dr. Kusemererwa,

I'm pleased to inform you that your manuscript has been deemed suitable for publication in PLOS One. Congratulations! Your manuscript is now being handed over to our production team.

Kind regards,

on behalf of

Dr Hamufare Dumisani Mugauri

Academic Editor

PLOS One